# ON RADEMACHER COMPLEXITY-BASED GENERALIZATION BOUNDS FOR DEEP LEARNING

## ABSTRACT

We show that the Rademacher complexity-based approach can generate non-vacuous generalisation bounds on Convolutional Neural Networks (CNNs) for classifying a small number of classes of images. The development of new contraction lemmas for high-dimensional mappings between vector spaces for general Lipschitz activation functions is a key technical contribution. These lemmas extend and improve the Talagrand contraction lemma in a variety of cases. Our generalisation bounds are based on the infinity norm of the weight matrices, distinguishing them from previous works that relied on different norms. Furthermore, while prior works that use the Rademacher complexity-based approach primarily focus on ReLU DNNs, our results extend to a broader class of activation functions.

## 1 INTRODUCTION

Deep models are typically heavily over-parametrized, while they still achieve good generalization performance. Despite the widespread use of neural networks in biotechnology, finance, health science, and business, just to name a selected few, the problem of understanding deep learning theoretically remains relatively under-explored. In 2002, Koltchinskii and Panchenko (Koltchinskii & Panchenko, 2002) proposed new probabilistic upper bounds on generalization error of the combination of many complex classifiers such as deep neural networks. These bounds were developed based on the general results of the theory of Gaussian, Rademacher, and empirical processes in terms of general functions of the margins, satisfying a Lipschitz condition. However, bounding Rademacher complexity for deep learning remains a challenging task. In this work, we present new upper bounds on the Rademacher complexity in deep learning, which differ from previous studies in how they depend on the norms of the weight matrices. Furthermore, we demonstrate that our bounds are non-vacuous for CNNs with a wide range of activation functions.

### 1.1 RELATED PAPERS

The complexity-based generalization bounds were established by traditional learning theory aiming to provide general theoretical guarantees for deep learning. (Goldberg & Jerrum, 1993), (Bartlett & Williamson, 1996), (Bartlett et al., 1998b) proposed upper bounds based on the VC dimension for DNNs. (Neyshabur et al., 2015) used Rademacher complexity to prove the bound with explicit exponential dependence on the network depth for ReLU networks. (Neyshabur et al., 2018) and (Bartlett et al., 2017) uses the PAC-Bayesian analysis and the covering number to obtain bounds with explicit polynomial dependence on the network depth, respectively. (Golowich et al., 2018) provided bounds with explicit square-root dependence on the depth for DNNs with positive-homogeneous activations such as ReLU.

The standard approach to develop generalization bounds on deep learning (and machine learning) was developed in seminar papers by (Vapnik, 1998), and it is based on bounding the difference between the generalization error and the training error. These bounds are expressed in terms of the so called VC-dimension of the class. However, these bounds are very loose when the VC-dimension of the class can be very large, or even infinite. In 1998, several authors (Bartlett et al., 1998a; Bartlett & Shawe-Taylor, 1999) suggested another class of upper bounds on generalization error that are expressed in terms of the empirical distribution of the margin of the predictor (the classifier). Later, Koltchinskii and Panchenko (Koltchinskii & Panchenko, 2002) proposed new probabilistic

upper bounds on the generalization error of the combination of many complex classifiers such as deep neural networks. These bounds were developed based on the general results of the theory of Gaussian, Rademacher, and empirical processes in terms of general functions of the margins, satisfying a Lipschitz condition. They improved previously known bounds on generalization error of convex combination of classifiers. Generalization bounds for deep learning and kernel learning with Markov dataset based on Rademacher and Gaussian complexity functions have recently analysed in (Truong, 2022a). Analysis of machine learning algorithms for Markov and Hidden Markov datasets already appeared in research literature (Duchi et al., 2011; Wang et al., 2019; Truong, 2022c).

In the context of supervised classification, PAC-Bayesian bounds have been used to explain the generalisation capability of learning algorithms (Langford & Shawe-Taylor, 2003; McAllester, 2004; A. Ambroladze & ShaweTaylor, 2007). Several recent works have focused on gradient descent based PAC-Bayesian algorithms, aiming to minimise a generalisation bound for stochastic classifiers (Dziugaite & Roy., 2017; W. Zhou & Orbanz., 2019; Biggs & Guedj, 2021). Most of these studies use a surrogate loss to avoid dealing with the zero-gradient of the misclassification loss. Several authors used other methods to estimate of the misclassification error with a non-zero gradient by proposing new training algorithms to evaluate the optimal output distribution in PAC-Bayesian bounds analytically (McAllester, 1998; Clerico et al., 2021b;a). Recently, (Nagarajan & Kolter, 2019) showed that uniform convergence might be unable to explain generalisation in deep learning by creating some examples where the test error is bounded by $\delta$ but the (two-sided) uniform convergence on this set of classifiers will yield only a vacuous generalisation guarantee larger than $1 - \delta$ for some $\delta \in (0, 1)$. There have been some interesting works which use information-theoretic approach to find PAC-bounds on generalization errors for machine learning (Xu & Raginsky, 2017; Esposito et al., 2021) and deep learning (Jakubovitz et al., 2018).

## 1.2 CONTRIBUTIONS

More specifically, our contributions are as follows:

- We develop new contraction lemmas for high-dimensional mappings between vector spaces which extend and improve the Talagrand contraction lemma for many cases.
- We apply our new contraction lemmas to each layer of a CNN.
- We validate our new theoretical results experimentally on CNNs for MNIST image classifications, and our bounds are non-vacuous when the number of classes is small.

As far as we know, this is the first result which shows that the Rademacher complexity-based approach can lead to non-vacuous generalisation bounds on CNNs.

## 1.3 OTHER NOTATIONS

Vectors and matrices are in boldface. For any vector $\mathbf{x} = (x_1, x_2, \cdots, x_n) \in \mathbb{R}^n$ where $\mathbb{R}$ is the field of real numbers, its induced-$L^p$ norm is defined as

$$\|\mathbf{x}\|_p = \left( \sum_{k=1}^{n} |x_k|^p \right)^{1/p}. \tag{1}$$

The $j$-th component of the vector $\mathbf{x}$ is denoted as $[\mathbf{x}]_j$ for all $j \in [n]$.

For $\mathbf{A} \in \mathbb{R}^{m \times n}$ where

$$\mathbf{A} = \begin{bmatrix} a_{11}, & a_{12}, & \cdots, & a_{1n} \\ a_{21}, & a_{22}, & \cdots, & a_{2n} \\ \vdots & \vdots & \ddots & \vdots \\ a_{m1}, & a_{m2}, & \cdots, & a_{mn} \end{bmatrix} \tag{2}$$

we defined the induced-norm of matrix $\mathbf{A}$ as

$$\|\mathbf{A}\|_{p,q} = \sup_{\mathbf{x} \neq \underline{0}} \frac{\|\mathbf{A}\mathbf{x}\|_q}{\|\mathbf{x}\|_p}. \tag{3}$$

For abbreviation, we also use the following notation

$$\|A\|_p := \|A\|_{p,p}. \tag{4}$$

It is known that

$$\|\mathbf{A}\|_1 = \max_{1 \le j \le n} \sum_{i=1}^m |a_{ij}|, \tag{5}$$

$$\|\mathbf{A}\|_2 = \sqrt{\lambda_{\max}(\mathbf{A}\mathbf{A}^T)}, \tag{6}$$

$$\|\mathbf{A}\|_\infty = \max_{1 \le i \le m} \sum_{j=1}^n |a_{ij}|, \tag{7}$$

where $\lambda_{\max}(\mathbf{A}\mathbf{A}^T)$ is defined as the maximum eigenvalue of the matrix $\mathbf{A}\mathbf{A}^T$ (or the square of the maximum singular value of $\mathbf{A}$).

## 2   CONTRACTION LEMMAS IN HIGH DIMENSIONAL VECTOR SPACES

First, we recall the Talagrand's contraction lemma.

**Lemma 1** *(Ledoux & Talagrand, 1991, Theorem 4.12) Let $\mathcal{H}$ be a hypothesis set of functions mapping from some set $\mathcal{X}$ to $\mathbb{R}$ and $\psi$ be a $\mu$-Lipschitz function from $\mathbb{R} \to \mathbb{R}$ for some $\mu > 0$. Then, for any sample $S$ of $n$ points $\mathbf{x}_1, \mathbf{x}_2, \cdots, \mathbf{x}_n \in \mathcal{X}$, the following inequality holds:*

$$\mathbb{E}_{\boldsymbol{\varepsilon}}\left[\sup_{h \in \mathcal{H}} \left|\frac{1}{n}\sum_{i=1}^n \varepsilon_i(\psi \circ h)(\mathbf{x}_i)\right|\right] \le 2\mu\mathbb{E}_{\boldsymbol{\varepsilon}}\left[\sup_{h \in \mathcal{H}} \left|\frac{1}{n}\sum_{i=1}^n \varepsilon_i h(\mathbf{x}_i)\right|\right], \tag{8}$$

*where $\boldsymbol{\varepsilon} = (\varepsilon_1, \varepsilon_2, \cdots, \varepsilon_n)$, and $\{\varepsilon_i\}_{i=1}^n$ is a sequence of i.i.d. Rademacher random variables (taking values $+1$ and $-1$ with probability $1/2$ each), independent of $\{\mathbf{x}_i\}$.*

In Theorem 2 below, we present a new version of Talagrand's contraction lemma for the high-dimensional mapping $\psi$ between vector spaces. The proof of the this theorem is provided in Appendix A.1 (Supplementary Material).

**Theorem 2** *Let $\mathcal{H}$ be a set of functions mapping from some set $\mathcal{X}$ to $\mathbb{R}^m$ for some $m \in \mathbb{Z}_+$ and*

$$\mathcal{L} = \left\{\psi_\alpha : \psi_\alpha(x) = ReLU(x) - \alpha ReLU(-x)\ \forall x \in \mathbb{R}, \alpha \in [0, 1]\right\} \tag{9}$$

*where $ReLU(x) = \max(x, 0)$.*

*For any $\mu > 0$, let $\psi : \mathbb{R} \to \mathbb{R}$ be a $\mu$-Lipschitz function. Define*

$$\mathcal{H}_+ = \begin{cases} \mathcal{H} \cup \{-h : h \in \mathcal{H}\}, & \text{if } \psi - \psi(0) \text{ is odd} \\ \mathcal{H} \cup \{-h : h \in \mathcal{H}\} \cup \{|h| : h \in \mathcal{H}\}, & \text{if } \psi - \psi(0) \text{ others} \end{cases}. \tag{10}$$

*Then, it holds that*

$$\mathbb{E}_{\boldsymbol{\varepsilon}}\left[\sup_{h \in \mathcal{H}} \left\|\frac{1}{n}\sum_{i=1}^n \varepsilon_i \psi(h(\mathbf{x}_i))\right\|_\infty\right]$$

$$\le \gamma(\mu)\mathbb{E}_{\boldsymbol{\varepsilon}}\left[\sup_{h \in \mathcal{H}_+} \left\|\frac{1}{n}\sum_{i=1}^n \varepsilon_i h(\mathbf{x}_i)\right\|_\infty\right] + \frac{1}{\sqrt{n}}|\psi(0)|, \tag{11}$$

*where*

$$\gamma(\mu) = \begin{cases} \mu, & \text{if } \psi - \psi(0) \text{ is odd or belongs to } \mathcal{L} \\ 2\mu, & \text{if } \psi - \psi(0) \text{ is even} \\ 3\mu, & \text{if } \psi - \psi(0) \text{ others} \end{cases}. \tag{12}$$

*Here, we define $\psi(\mathbf{x}) := (\psi(x_1), \psi(x_2), \cdots, \psi(x_m))^T$ for any $\mathbf{x} = (x_1, x_2, \cdots, x_m)^T \in \mathbb{R}^m$.*

**Remark 3** *Some remarks are in order.*

- *Identity, ReLU, Leaky ReLU, Parametric rectified linear unit (PReLU) belong to the class of functions $\mathcal{L}$.*

- *If $\psi$ is odd or belongs to $\mathcal{L}$, then $\psi(0) = 0$. Therefore, Theorem 2 improves Lemma 1 in the special case where $m = 1$. This enhancement is achieved by leveraging the unique properties of certain function classes.*

- *Our results are based on a novel approach, which shows that tighter contraction lemmas can be obtained when both the class of functions $\mathcal{H}$ and the activation functions possess certain special properties. More specifically, in this work, we extend the class of functions $\mathcal{H}$ by adding more functions, resulting in a new class $\mathcal{H}_+$, which possesses certain special properties. Additionally, we restrict the class of activation functions to $\mathcal{L} \cup \{\psi : \mathbb{R} \to \mathbb{R} : \psi(x) - \psi(0) = -(\psi(-x) - \psi(0)), \ \ \forall x \in \mathbb{R}\}$.*

Now, the following result can be easily proved (See Appendix A.6 in Supplementary Material).

**Theorem 4** *Let $\mathcal{G}$ be a class of functions from $\mathbb{R}^r \to \mathbb{R}^q$ and $\mathcal{V}$ be a class of matrices $\mathbf{W}$ on $\mathbb{R}^{p \times q}$ such that $\sup_{\mathbf{W} \in \mathcal{V}} \|\mathbf{W}\|_\infty \leq \nu$. Then, it holds that*

$$\mathbb{E}_{\boldsymbol{\varepsilon}} \left[ \sup_{\mathbf{W} \in \mathcal{V}} \sup_{f \in \mathcal{G}} \left\| \frac{1}{n} \sum_{i=1}^n \varepsilon_i \mathbf{W} f(\mathbf{x}_i) \right\|_\infty \right] \leq \nu \mathbb{E}_{\boldsymbol{\varepsilon}} \left[ \sup_{f \in \mathcal{G}} \left\| \frac{1}{n} \sum_{i=1}^n \varepsilon_i f(\mathbf{x}_i) \right\|_\infty \right]. \tag{13}$$

## 3 RADEMACHER COMPLEXITY BOUNDS FOR CONVOLUTIONAL NEURAL NETWORKS (CNNS)

### 3.1 CONVOLUTIONAL NEURAL NETWORK MODELS

Let $d_0, d_1, \cdots, d_L, d_{L+1}$ be a sequence of positive integer numbers such that $d_0 = d$ for some fixed $d \in \mathbb{Z}_+$. We define a class of function $\mathcal{F}$ as follows:

$$\mathcal{F} := \left\{ f = f_L \circ f_{L-1} \circ \cdots \circ f_1 \circ f_0 : f_i \in \mathcal{G}_i \subset \{g_i : \mathbb{R}^{d_i} \to \mathbb{R}^{d_{i+1}}\}, \quad \forall i \in \{1, 2, \cdots, L\} \right\}, \tag{14}$$

where $f_0 : [0,1]^d \to \mathbb{R}^{d_1}$ is a fixed function and $d_{i+1} = M$ for some $M \in \mathbb{Z}_+$. A Convolutional Neural Network (CNN) with network-depth $L$ is defined as a composition map $f \in \mathcal{F}$ where

$$f_i(\mathbf{x}) = \sigma_i(\mathbf{W}_i \mathbf{x}), \quad \forall \mathbf{x} \in \mathbb{R}^{d_i}. \tag{15}$$

Here, $\mathbf{W}_i \in \mathcal{W}_i$ where $\mathcal{W}_i$ is a set of matrices in $\mathbb{R}^{d_{i+1} \times d_i}$, and $\sigma_i$ is a mapping from $\mathbb{R}^{d_{i+1}} \to \mathbb{R}^{d_{i+1}}$.

Given a function $f \in \mathcal{F}$, a function $g \in \mathbb{R}^M \times [M]$ predicts a label $y \in [M]$ for an example $\mathbf{x} \in \mathbb{R}^d$ if and only if

$$g(f(\mathbf{x}), y) > \max_{y' \neq y} g(f(\mathbf{x}), y') \tag{16}$$

where $g(f(\mathbf{x}), y) = \mathbf{w}_y^T f(\mathbf{x})$ with $\mathbf{w}_y = \underbrace{(0, 0, \cdots, 0, 1, 0, \cdots, 0)}_{\mathbf{w}_y(y)=1}$.

For a training set $\{\mathbf{x}_i\}_{i=1}^n$, the $\infty$-norm *Rademacher complexity* for the class function $\mathcal{F}$ is defined as

$$R_n(\mathcal{F}) := \mathbb{E}_{\boldsymbol{\varepsilon}} \left[ \sup_{f \in \mathcal{F}} \left\| \frac{1}{n} \sum_{i=1}^n \varepsilon_i f(\mathbf{x}_i) \right\|_\infty \right]. \tag{17}$$

### 3.2 SOME CONTRACTION LEMMAS FOR CNNS

Based on Theorem 2 and Theorem 4, the following versions of Talagrand's contraction lemma for different layers of CNN are derived.

**Definition 5 (Convolutional Layer with Average Pooling)** *Let $\mathcal{G}$ be a class of $\mu$-Lipschitz function $\sigma$ from $\mathbb{R} \to \mathbb{R}$ such that $\sigma(0)$ is fixed. Let $C, Q \in \mathbb{Z}_+$, $\{r_l, \tau_l\}_{l \in [Q]}$ be two tuples of positive integer numbers, and $\{W_{l,c} \in \mathbb{R}^{r_l \times r_l}, c \in [C], l \in [Q]\}$ be a set of kernel matrices. A convolutional layer with average pooling, $C$ input channels, and $Q$ output channels is defined as a set of $Q \times C$ mappings $\Psi = \{\psi_{l,c}, l \in [Q], c \in [C]\}$ from $\mathbb{R}^{d \times d}$ to $\mathbb{R}^{\lceil (d-r_l+1)/\tau_l \rceil \times \lceil (d-r_l+1)/\tau_l \rceil}$ such that*

$$\psi_{l,c}(\mathbf{x}) = \sigma_{\mathrm{avg}} \circ \sigma_{l,c}(\mathbf{x}), \tag{18}$$

*where*

$$\sigma_{\mathrm{avg}}(\mathbf{x}) = \frac{1}{\tau_l^2} \Big( \sum_{k=1}^{\tau_l^2} x_k, \cdots, \sum_{k=(j-1)\tau_l^2+1}^{j\tau_l^2} x_k, \cdots, \sum_{k=\lceil (d-r_l+1)^2/\tau_l^2 \rceil - r_l^2 + 1}^{\lceil (d-r_l+1)^2/\tau_l^2 \rceil \tau_l^2} x_k \Big),$$

$$\forall \mathbf{x} \in \mathbb{R}^{\lceil (d-r_l+1)^2/\tau_l^2 \rceil \tau_l^2}, \tag{19}$$

*and for all $\mathbf{x} \in \mathbb{R}^{d \times d \times C}$,*

$$\sigma_{l,c}(\mathbf{x}) = \{\hat{x}_c(a,b)\}_{a,b=1}^{d-r_l+1}, \tag{20}$$

$$\hat{x}_c(a,b) = \sigma\Big( \sum_{u=0}^{r_l-1} \sum_{v=0}^{r_l-1} x(a+u, b+v, c) W_{l,c}(u+1, v+1) \Big). \tag{21}$$

**Lemma 6 (Convolutional Layer with Average Pooling)** *Let $\mathcal{F}$ be a set of functions mapping from some set $\mathcal{X}$ to $\mathbb{R}^m$ for some $m \in \mathbb{Z}_+$. Consider a convolutional layer with average pooling defined in Definition 5. Recall the definition of $\mathcal{L}$ in (9). Then, it hold that*

$$\mathbb{E}_{\boldsymbol{\varepsilon}} \left[ \sup_{c \in [C]} \sup_{l \in [Q]} \sup_{\psi_l \in \Psi} \sup_{f \in \mathcal{F}} \left\| \frac{1}{n} \sum_{i=1}^n \varepsilon_i \psi_{l,c} \circ f(\mathbf{x}_i) \right\|_\infty \right]$$

$$\leq \left[ \gamma(\mu) \sup_{c \in [C]} \sup_{l \in [Q]} \Big( \sum_{u=0}^{r_l-1} \sum_{v=0}^{r_l-1} |W_{l,c}(u+1, v+1)| \Big) \right] \mathbb{E} \left[ \sup_{f \in \mathcal{F}_+} \left\| \frac{1}{n} \sum_{i=1}^n \varepsilon_i f(\mathbf{x}_i) \right\|_\infty \right] + \frac{|\sigma(0)|}{\sqrt{n}}, \tag{22}$$

*where*

$$\gamma(\mu) = \begin{cases} \mu, & \text{if } \sigma - \sigma(0) \text{ is odd or belongs to } \mathcal{L} \\ 2\mu, & \text{if } \sigma - \sigma(0) \text{ is even} \\ 3\mu, & \text{if } \sigma - \sigma(0) \text{ others} \end{cases}. \tag{23}$$

*Here,*

$$\mathcal{F}_+ = \begin{cases} \mathcal{F} \cup \{-f : f \in \mathcal{F}\}, & \text{if } \sigma - \sigma(0) \text{ is odd} \\ \mathcal{F} \cup \{-f : f \in \mathcal{F}\} \cup \{|f| : f \in \mathcal{F}\}, & \text{if } \sigma - \sigma(0) \text{ others} \end{cases}. \tag{24}$$

For Dropout layer, the following holds:

**Lemma 7 (Dropout Layers)** *Let $\psi(\mathbf{x})$ is the output of the $\mathbf{x}$ via the Dropout layer. Then, it holds that*

$$\mathbb{E}_{\boldsymbol{\varepsilon}} \left[ \sup_{f \in \mathcal{H}} \left\| \frac{1}{n} \sum_{i=1}^n \varepsilon_i \psi \circ f(\mathbf{x}_i) \right\|_\infty \right] \leq \mathbb{E} \left[ \sup_{f \in \mathcal{H}} \left\| \frac{1}{n} \sum_{i=1}^n \varepsilon_i f(\mathbf{x}_i) \right\|_\infty \right]. \tag{25}$$

The following Rademacher complexity bounds for Dense Layers.

**Lemma 8 (Dense Layers)** *Recall the definition of $\mathcal{L}$ in (9). Let $\mathcal{G}$ be a class of $\mu$-Lipschitz function, i.e.,*

$$\big| \sigma(x) - \sigma(y) \big| \leq \mu |x - y|, \qquad \forall x, y \in \mathbb{R}, \tag{26}$$

*such that $\sigma(0)$ is fixed. Let $\mathcal{V}$ be a class of matrices $\mathbf{W}$ on $\mathbb{R}^{d\times d'}$ such that $\sup_{\mathbf{W}\in\mathcal{V}}\|\mathbf{W}\|_\infty \leq \beta$. For any vector $\mathbf{x} = (x_1, x_2, \cdots, x_{d'})$, we denote by $\sigma(\mathbf{x}) := (\sigma(x_1), \sigma(x_2), \cdots, \sigma(x_{d'}))^T$. Then, it holds that*

$$\mathbb{E}_{\boldsymbol{\varepsilon}}\left[\sup_{\mathbf{W}\in\mathcal{V}}\sup_{f\in\mathcal{G}}\left\|\frac{1}{n}\sum_{i=1}^n \varepsilon_i\sigma(\mathbf{W}f(\mathbf{x}_i))\right\|_\infty\right]$$

$$\leq \gamma(\mu)\beta\mathbb{E}_{\boldsymbol{\varepsilon}}\left[\sup_{f\in\mathcal{G}}\left\|\frac{1}{n}\sum_{i=1}^n \varepsilon_i f(\mathbf{x}_i)\right\|_\infty\right] + \frac{|\sigma(0)|}{\sqrt{n}}, \tag{27}$$

*where*

$$\gamma(\mu) = \begin{cases} \mu, & \text{if } \sigma - \sigma(0) \text{ is odd or belongs to } \mathcal{L} \\ 2\mu, & \text{if } \sigma - \sigma(0) \text{ is even} \\ 3\mu, & \text{if } \sigma - \sigma(0) \text{ others} \end{cases}. \tag{28}$$

**Remark 9** *The convolutional layer with average pooling, dropout layers, and dense layers can be viewed as compositions of linear mappings and pointwise activation functions. Therefore, Lemmas 6, 7, and 8 are derived by applying Theorem 2 to the pointwise mappings and Theorem 4 to the linear mappings.*

### 3.3 RADEMACHER COMPLEXITY BOUNDS FOR CNNs

In this section, we show the following result.

**Theorem 10** *Let*

$$\mathcal{L} = \left\{\psi_\alpha : \psi_\alpha(x) = ReLU(x) - \alpha ReLU(-x) \ \forall x \in \mathbb{R}, \alpha \in [0,1]\right\}. \tag{29}$$

*Consider the CNN defined in Section 3.1 where*

$$[f_i(\mathbf{x})]_j = \sigma_i\big(\mathbf{w}_{j,i}^T f_{i-1}(\mathbf{x})\big) \ \forall j \in [d_{i+1}]$$

*and $\sigma_i$ is $\mu_i$-Lipschitz. In addition, $f_0(\mathbf{x}) = [\mathbf{x}^T, 1]^T$, $\forall \mathbf{x} \in \mathbb{R}^d$ and $\mathbf{x}$ is normalised such that $\|\mathbf{x}\|_\infty \leq 1$. Let*

$$\mathcal{K} = \{i \in [L] : layer \ i \ is \ a \ convolutional \ layer \ with \ average \ pooling\}, \tag{30}$$

$$\mathcal{D} = \{i \in [L] : layer \ i \ is \ a \ dropout \ layer\}. \tag{31}$$

*We assume that there are $Q_i$ kernel matrices $W_i^{(l)}$'s of size $r_i^{(l)} \times r_i^{(l)}$ for the $i$-th convolutional layer. For all the (dense) layers that are not convolutional, we define $\mathbf{W}_i$ as their coefficient matrices. In addition, define*

$$\gamma_{\text{cvl,i}} = \gamma(\mu_i) \sup_{l\in[Q_i]} \sum_{u=1}^{r_{i,l}} \sum_{v=1}^{r_{i,l}} |W_i^{(l)}(u,v)|, \tag{32}$$

$$\gamma_{\text{dl,i}} = \gamma(\mu_i)\big\|\mathbf{W}_i\big\|_\infty \quad i \notin \mathcal{K}. \tag{33}$$

*where*

$$\gamma(\mu_i) = \begin{cases} \mu_i, & \text{if } \sigma_i - \sigma_i(0) \text{ is odd or belongs to } \mathcal{L} \\ 2\mu, & \text{if } \sigma_i - \sigma_i(0) \text{ is even} \\ 3\mu, & \text{if } \sigma_i - \sigma_i(0) \text{ others} \end{cases}. \tag{34}$$

*Then, the Rademacher complexity, $\mathcal{R}_n(\mathcal{F})$, satisfies*

$$\mathcal{R}_n(\mathcal{F}) := \mathbb{E}_{\boldsymbol{\varepsilon}}\left[\sup_{f\in\mathcal{F}_+}\left\|\frac{1}{n}\sum_{i=1}^n \varepsilon_i f(\mathbf{x}_i)\right\|_\infty\right]$$

$$\leq F_L, \tag{35}$$

*where $F_L$ is estimated by the following recursive expression:*

$$F_i = \begin{cases} F_{i-1}\gamma_{\text{cvl,i}} + \frac{|\sigma_i(0)|}{\sqrt{n}}, & i \in \mathcal{K} \\ F_{i-1}\gamma_{\text{dl,i}} + \frac{|\sigma_i(0)|}{\sqrt{n}}, & i \notin (\mathcal{K}\cup\mathcal{D}) \\ F_{i-1}, & i \in \mathcal{D} \end{cases} \tag{36}$$

*and $F_0 = \sqrt{\frac{d+1}{n}}$.*

**Proof** This is a direct application of Lemmas 6, 7, and 8. By the modelling of CNNs in Section 3.1, it holds that

$$\mathcal{F}_k := \left\{ f = f_k \circ f_{k-1} \circ \cdots \circ f_1 \circ f_0 : f_i \in \mathcal{G}_i \subset \{g_i : \mathbb{R}^{d_i} \to \mathbb{R}^{d_{i+1}}\}, \quad \forall i \in \{1, 2, \cdots, k\} \right\} \tag{37}$$

and $\mathcal{F} := \mathcal{F}_L$.

For CNNs, $f_l(\mathbf{x}) = \sigma_l(W_l \mathbf{x})$ for all $l \in [L]$ where $W_l \in \mathcal{W}_l$ (a set of matrices) and $\sigma_l \in \Psi_l$ where

$$\Psi_l = \left\{ \sigma_l : |\sigma_l(x) - \sigma_l(y)| \le \mu_l |x - y|, \quad \forall x, y \in \mathbb{R} \right\}. \tag{38}$$

Then, since $|\sigma_l|, -\sigma_l \in \Psi_l$, it is easy to see that

$$\mathcal{F}_{l,+} \subset \Psi_l(\mathcal{W}_l \mathcal{F}_{l-1,+}), \qquad \forall l \in [L], \tag{39}$$

where $\mathcal{F}_{l,+}$ is a supplement of $\mathcal{F}_l$ defined in (24).

Therefore, by peeling layer by layer we finally have

$$\mathbb{E}_{\boldsymbol{\varepsilon}} \left[ \sup_{f \in \mathcal{F}} \left\| \frac{1}{n} \sum_{i=1}^n \varepsilon_i f(\mathbf{x}_i) \right\|_\infty \right] \le F_L, \tag{40}$$

where for each $i \in [L]$

$$F_i = \begin{cases} F_{i-1} \gamma_{\text{cvl,i}} + \frac{|\sigma_i(0)|}{\sqrt{n}}, & i \in \mathcal{K} \\ F_{i-1} \gamma_{\text{dl,i}} + \frac{|\sigma_i(0)|}{\sqrt{n}}, & i \notin (\mathcal{K} \cup \mathcal{D}) \\ F_{i-1}, & i \in \mathcal{D} \end{cases} \tag{41}$$

and

$$F_0 = \mathbb{E}_{\boldsymbol{\varepsilon}} \left[ \sup_{f \in \mathcal{H}_+} \left\| \frac{1}{n} \sum_{i=1}^n \varepsilon_i f(\mathbf{x}_i) \right\|_\infty \right]. \tag{42}$$

Here, $\mathcal{H}_+$ is the extended set of inputs to the CNN, i.e.,

$$\mathcal{H}_+ = \begin{cases} f_0 \cup \{-f_0\}, & \text{if } \sigma_1 - \sigma_1(0) \text{ is odd} \\ f_0 \cup \{-f_0\} \cup \{|f_0|\}, & \text{if } \sigma_1 - \sigma_1(0) \text{ others} \end{cases}. \tag{43}$$

Now, for the case $\sigma_1 - \sigma_1(0)$ is odd, it is easy to see that

$$\sup_{f \in \mathcal{H}_+} \left\| \frac{1}{n} \sum_{i=1}^n \varepsilon_i f(\mathbf{x}_i) \right\|_\infty = \left\| \frac{1}{n} \sum_{i=1}^n \varepsilon_i f_0(\mathbf{x}_i) \right\|_\infty \tag{44}$$

$$\le \left\| \frac{1}{n} \sum_{i=1}^n \varepsilon_i f_0(\mathbf{x}_i) \right\|_2. \tag{45}$$

On the other hand, for the case $\sigma_1 - \sigma_1(0)$ is general, we have

$$\sup_{f \in \mathcal{H}_+} \left\| \frac{1}{n} \sum_{i=1}^n \varepsilon_i f(\mathbf{x}_i) \right\|_\infty \le \max \left\{ \left\| \frac{1}{n} \sum_{i=1}^n \varepsilon_i f_0(\mathbf{x}_i) \right\|_\infty, \left\| \frac{1}{n} \sum_{i=1}^n \varepsilon_i |f_0(\mathbf{x}_i)| \right\|_\infty \right\}. \tag{46}$$

On the other hand, we have

$$\mathbb{E}_{\boldsymbol{\varepsilon}} \left[ \left\| \frac{1}{n} \sum_{i=1}^n \varepsilon_i f_0(\mathbf{x}_i) \right\|_2 \right]$$

$$\le \frac{1}{n} \sqrt{ \mathbb{E}_{\boldsymbol{\varepsilon}} \left[ \left\| \frac{1}{n} \sum_{i=1}^n \varepsilon_i f_0(\mathbf{x}_i) \right\|_2^2 \right]} \tag{47}$$

$$\le \frac{1}{n} \sqrt{ \sum_{j=1}^{d+1} \sum_{i=1}^n [f_0(\mathbf{x}_i)]_j^2} \tag{48}$$

$$\le \frac{1}{n} \sqrt{(d+1)n} \tag{49}$$

$$= \sqrt{\frac{d+1}{n}}, \tag{50}$$

where (49) follows from $|[f_0(\mathbf{x}_i)]_j| \leq 1$ for all $i \in [n], j \in [d_1]$ when the data is normalised by using the standard method.

Similarly, we also have

$$\mathbb{E}_{\boldsymbol{\varepsilon}}\left[\left\|\frac{1}{n}\sum_{i=1}^{n}\varepsilon_i|f_0(\mathbf{x}_i)|\right\|_2\right] \leq \sqrt{\frac{d+1}{n}}. \tag{51}$$

# 4 GENERALIZATION BOUNDS FOR CNNs

## 4.1 GENERALIZATION BOUNDS FOR DEEP LEARNING

**Definition 11** *Recall the CNN model in Section 3.1. The margin of a labelled example $(\mathbf{x}, y)$ is defined as*

$$m_f(\mathbf{x}, y) := g(f(\mathbf{x}), y) - \max_{y' \neq y} g(f(\mathbf{x}), y'), \tag{52}$$

*so $f$ mis-classifies the labelled example $(\mathbf{x}, y)$ if and only if $m_f(\mathbf{x}, y) \leq 0$. The generalisation error is defined as $\mathbb{P}(m_f(\mathbf{x}, y) \leq 0)$. It is easy to see that $\mathbb{P}(m_f(\mathbf{x}, y) \leq 0) = \mathbb{P}(\mathbf{w}_y^T f(\mathbf{x}) \leq \max_{y' \in \mathcal{Y}} \mathbf{w}_{y'}^T f(\mathbf{x}))$.*

**Remark 12** *Some remarks:*

- *Since $g(f(\mathbf{x}), y) > \max_{y' \neq y} g(f(\mathbf{x}), y')$, it holds that $\tilde{g}(f_k(\mathbf{x}, y)) > \max_{y' \neq y} \tilde{g}(f_k(\mathbf{x}, y'))$ for some $k \in [L]$ where $\tilde{g}$ is an arbitrary function. Hence, $\mathbb{P}(m_f(\mathbf{x}, y) \leq 0) \leq \mathbb{P}(\tilde{g}(f_k(\mathbf{x}, y)) > \max_{y' \neq y} \tilde{g}(f_k(\mathbf{x}, y')))$, so we can bound the generalisation error by using only a part of CNN networks (from layer $0$ to layer $k$). However, we need to know $\tilde{g}$. If the last layers of CNN are softmax, we can easily know this function.*

- *When testing on CNNs, it usually happens that the generalisation error bound becomes smaller when we use almost all layers.*

Now, we prove the following lemma.

**Lemma 13** *Let $\mathcal{F}$ be a class of function from $\mathcal{X}$ to $\mathbb{R}^m$. For CNNs for classification, it holds that*

$$\mathbb{E}_{\boldsymbol{\varepsilon}}\left[\sup_{f \in \mathcal{F}}\left|\frac{1}{n}\sum_{i=1}^{n}\varepsilon_i m_f(\mathbf{x}_i, y_i)\right|\right] \leq \beta(M)\mathbb{E}_{\boldsymbol{\varepsilon}}\left[\sup_{f \in \mathcal{F}}\left\|\frac{1}{n}\sum_{i=1}^{n}\varepsilon_i m_f(\mathbf{x}_i)\right\|_\infty\right], \tag{53}$$

*where*

$$\beta(M) = \begin{cases} M(2M-1), & M > 2 \\ 2M, & M = 2 \end{cases}. \tag{54}$$

For $M > 2$, (53) is a result of (Koltchinskii & Panchenko, 2002, Proof of Theorem 11). We improve this constant for $M = 2$. Based on the above Rademacher complexity bounds and a justified application of McDiarmid's inequality, we obtains the following generalization for deep learning with i.i.d. datasets.

**Theorem 14** *Let $\gamma > 0$ and define the following function (the $\gamma$-margin cost):*

$$\zeta(x) := \begin{cases} 0, & \gamma \leq x \\ 1 - x/\gamma, & 0 \leq x \leq \gamma \\ 1, & x \leq 0 \end{cases}. \tag{55}$$

*Recall the definition of the average Rademacher complexity $\mathcal{R}_n(\mathcal{F})$ in (35) and the definition of $\beta(M)$ in (54). Let $\{(\mathbf{x}_i, y_i)\}_{i=1}^n \sim P_{\mathbf{x}y}$ for some joint distribution $P_{\mathbf{x}y}$ on $\mathcal{X} \times \mathcal{Y}$. Then, for any $t > 0$, the following holds:*

$$\mathbb{P}\left\{\exists f \in \mathcal{F} : \mathbb{P}(m_f(\mathbf{x}, y) \leq 0) > \inf_{\gamma \in (0,1]}\left[\frac{1}{n}\sum_{i=1}^{n}\zeta(m_f(\mathbf{x}_i, y_i))\right.\right.$$

$$\left.\left. + \frac{2\beta(M)}{\gamma}\mathcal{R}_n(\mathcal{F}) + \frac{2t + \sqrt{\log\log_2(2\gamma^{-1})}}{\sqrt{n}}\right]\right\} \leq 2\exp(-2t^2). \tag{56}$$

**Corollary 15** *(PAC-bound) Recall the definition of the average Rademacher complexity $\mathcal{R}_n(\mathcal{F})$ in* (35) *and the definition of $\beta(M)$ in* (54). *Let $\{(\mathbf{x}_i, y_i)\}_{i=1}^n \sim P_{\mathbf{x}y}$ for some joint distribution $P_{\mathbf{x}y}$ on $\mathcal{X} \times \mathcal{Y}$. Then, for any $\delta \in (0,1]$, with probability at least $1 - \delta$, it holds that*

$$\mathbb{P}\big(m_f(\mathbf{x}, y) \leq 0\big) \leq \inf_{\gamma \in (0,1]} \left[ \frac{1}{n} \sum_{i=1}^n \mathbf{1}\big\{m_f(\mathbf{x}_i, y_i) \leq \gamma\big\} \right.$$

$$\left. + \frac{2\beta(M)}{\gamma} \mathcal{R}_n(\mathcal{F}) + \sqrt{\frac{\log \log_2(2\gamma^{-1})}{n}} + \sqrt{\frac{2}{n} \log \frac{3}{\delta}} \right], \qquad \forall f \in \mathcal{F}. \tag{57}$$

**Proof** This result is obtain from Theorem 14 by choosing $t > 0$ such that $3\exp(-2t^2) = \delta$.

## 5 NUMERICAL RESULTS

In this experiment, we use a CNN (cf. Fig. 1) for classifying MNIST images (class 0 and class 1), i.e., $M = 2$, which consists of $n = 12665$ training examples.

For this model, the sigmoid activation $\sigma$ satisfies $\sigma(x) - \sigma(0) = \frac{1}{2}\tanh\left(\frac{x}{2}\right)$ which is odd and has the Lipschitz constant $1/4$. In addition, for the dense layer, the sigmoid activation satisfies

$$\big|\sigma(x) - \sigma(y)\big| \leq \frac{1}{4}|x - y|, \qquad \forall x, y \in \mathbb{R}. \tag{58}$$

Hence, by Theorem 10 it holds that $\mathcal{R}_n(\mathcal{F}) \leq F_3$, where

$$F_3 \leq \underbrace{\frac{1}{4}\|\mathbf{W}\|_\infty F_2 + \frac{1}{2\sqrt{n}}}_{\text{Dense layer}}, \tag{59}$$

$$F_2 \leq \underbrace{\left(\frac{1}{4} \sup_{l \in [64]} \sum_{u=1}^3 \sum_{v=1}^3 \big|W_2^{(l)}(u,v)\big|\right) F_1 + \frac{1}{2\sqrt{n}}}_{\text{The second convolutional layer}}, \tag{60}$$

$$F_1 \leq \underbrace{\left(\frac{1}{4} \sup_{l \in [32]} \sum_{u=1}^3 \sum_{v=1}^3 \big|W_1^{(l)}(u,v)\big|\right) F_0 + \frac{1}{2\sqrt{n}}}_{\text{The first convolutional layer}}, \tag{61}$$

$$F_0 = \sqrt{\frac{d+1}{n}}. \tag{62}$$

Numerical estimation of $F_3$ gives $\mathcal{R}_n(\mathcal{F}) \leq 0.00859$.

By Corollary 15 with probability at least $1 - \delta$, it holds that

$$\mathbb{P}\big(m_f(\mathbf{x}, y) \leq 0\big) \leq \inf_{\gamma \in (0,1]} \left[ \frac{1}{n} \sum_{i=1}^n \zeta\big(m_f(\mathbf{x}_i, y_i)\big) \right.$$

$$\left. + \frac{4M}{\gamma} \mathcal{R}_n(\mathcal{F}) + \sqrt{\frac{\log \log_2(2\gamma^{-1})}{n}} + \sqrt{\frac{2}{n} \log \frac{3}{\delta}} \right] \tag{63}$$

By setting $\delta = 5\%$, $\gamma = 0.5$, the generalisation error can be upper bounded by

$$\mathbb{P}\big(m_f(\mathbf{x}, y) \leq 0\big) \leq 0.189492. \tag{64}$$

For this model, the reported test error is $0.0028368$.

Two extra experiments are given in Supplementary Materials.

```python
model = keras.Sequential(
    [
        keras.Input(shape=input_shape),
        layers.Conv2D(32, kernel_size=(3, 3), activation="sigmoid"),
        layers.AveragePooling2D(pool_size=(2, 2)),
        layers.Conv2D(64, kernel_size=(3, 3), activation="sigmoid"),
        layers.AveragePooling2D(pool_size=(2, 2)),
        layers.Flatten(),
        layers.Dropout(0.5),
        layers.Dense(2, activation="sigmoid"),
    ]
)
```

Figure 1: CNN model with sigmoid activations

# 6 COMPARISION WITH GOLOWICH ET AL.'S BOUND (GOLOWICH ET AL., 2018)

In (Golowich et al., 2018, Section 4), the authors present an upper bound on Rademacher complexity for DNNs with ReLU activation functions as follows:

$$\mathcal{R}_n(\mathcal{F}) = O\bigg( \prod_{j=1}^{L} \|\mathbf{W}_j\|_F \max\bigg\{ 1, \log\bigg( \prod_{j=1}^{L} \frac{\|\mathbf{W}_j\|_F}{\|\mathbf{W}_j\|_2} \bigg) \bigg\} \min\bigg\{ \frac{\max\{1, \log n\}^{3/4}}{n^{1/4}}, \sqrt{\frac{L}{n}} \bigg\} \bigg)$$

(65)

where $\mathbf{W}_1, \mathbf{W}_2, \cdots, \mathbf{W}_L$ are the parameter matrices of the $L$ layers.

Now, let $\Gamma$ be the term inside the bracket in (65), and define

$$\beta = \min_j \frac{\|\mathbf{W}_j\|_F}{\|\mathbf{W}_j\|_2} \geq 1.$$

(66)

Then, from (65) we have

$$\Gamma \geq \prod_{j=1}^{L} \|\mathbf{W}_j\|_F \min\bigg\{ \frac{\max\{1, \log n\}^{3/4} \sqrt{\max\{1, L\log\beta\}}}{n^{1/4}}, \sqrt{\frac{L}{n}} \bigg\}.$$

(67)

For the general case, it holds that $\beta > 1$. Hence, from (67) we have

$$\mathcal{R}_n(\mathcal{F}) = O\bigg( \sqrt{\frac{L}{n}} \prod_{j=1}^{L} \|\mathbf{W}_j\|_F \bigg).$$

(68)

As analysed in (Golowich et al., 2018), this bound improves many previous bounds, including Neyshabur et al.'s bound Neyshabur et al. (2015), Neyshabur et al. (2018) which are known to be vacuous for certain ReLU DNNs (Nagarajan & Kolter, 2019).

By using Theorem 10 and Lemma 8, we can show that

$$\mathcal{R}_n(\mathcal{F}) = O\bigg( \sqrt{\frac{1}{n}} \prod_{j=1}^{L} \mu_j \|\mathbf{W}_j\|_\infty \bigg)$$

(69)

for DNNs with some special classes of activation functions, including ReLU family and classes of old activation functions, where $\mu_j$ is the Lipschitz constant of the $j$-layer activation function.

In general, the Frobenius norm $\|\mathbf{W}_j\|_F$ of $\mathbf{W}_j$ can be either larger or smaller than its infinity norm $\|\mathbf{W}_j\|_\infty$, depending on the specific case. For example, suppose that $\mathbf{W}_j$ is a sparse matrix with only one non-zero element $a_k$ in the $k$-row, for all $k \in [d_{j+1}]$. Then, we have $\|\mathbf{W}_j\|_F = \sqrt{\sum_{k=1}^{d_{j+1}} |a_k|^2} \geq \max_{1 \leq k \leq d_{j+1}} |a_k| = \|\mathbf{W}_j\|_\infty$. Hence, (69) provides a new way to characterize the generalisation error in ReLU DNNs, which differ from previous studies in how they depend on the norms of the weight matrices.

Additionally, our bound in (69) is applicable to a broad range of activation functions. While ReLU DNNs are primarily considered in the works of (Golowich et al., 2018), Neyshabur et al. (2015), and Neyshabur et al. (2018), our approach extends to many other activation functions as well.

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
