# ON RADEMACHER COMPLEXITY-BASED GENERALIZATION BOUNDS FOR DEEP LEARNING

## ABSTRACT

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

# A APPENDIX

## A.1 PROOF OF THEOREM 2

The proof of Theorem 2 is a combination of the following contraction lemmas.

**Lemma 16** *Let $\mathcal{H}$ be a set of functions mapping $\mathcal{X}$ to $\mathbb{R}^m$ and $\mathcal{H}_+ = \mathcal{H} \cup \{|h| : h \in \mathcal{H}\}$ and $\psi : \mathbb{R} \to \mathbb{R}$ such that $\psi(x) = ReLU(x) - \alpha ReLU(-x) \ \forall x$ for some $\alpha \in [0,1]$. Then, for any $p \geq 1$ it holds that*

$$\mathbb{E}_{\boldsymbol{\varepsilon}}\left[\sup_{h \in \mathcal{H}} \left\|\frac{1}{n}\sum_{i=1}^{n}\varepsilon_i \psi(h(\mathbf{x}_i))\right\|_p\right] \leq \mathbb{E}_{\boldsymbol{\varepsilon}}\left[\sup_{h \in \mathcal{H}_+} \left\|\frac{1}{n}\sum_{i=1}^{n}\varepsilon_i h(\mathbf{x}_n)\right\|_p\right]. \tag{69}$$

Identity, ReLU, Leaky ReLU, Parametric rectified linear unit (PReLU) belong to the class of functions $\mathcal{L} := \{\psi : \psi(x) = ReLU(x) - \alpha ReLU(-x) \ \forall x, \ \text{for some} \ \alpha \in \mathbb{R}\}$.

**Lemma 17** *Let $\mathcal{H}$ be a set of functions mapping $\mathcal{X}$ to $\mathbb{R}^m$. Define*

$$\mathcal{H}_+ = \mathcal{H} \cup \big\{ - h : h \in \mathcal{H}\big\}. \tag{70}$$

*For any $\mu > 0$, let $\psi : \mathbb{R}^m \to \mathbb{R}^m$ such that $\big|\psi_j(\mathbf{x}) - \psi_j(\mathbf{x}')\big| \leq \mu|x_j - x_j'|, \ \forall(\mathbf{x}, \mathbf{x}') \in \mathbb{R}^m \times \mathbb{R}^m\}, \forall j \in [m]$ and $\psi - \psi(\mathit{0})$ is odd. In addition, $\psi_j(\mathit{0})$ does not depend on $j$. Then, it holds that*

$$\mathbb{E}_{\boldsymbol{\varepsilon}}\left[\sup_{h \in \mathcal{H}} \left\|\frac{1}{n}\sum_{i=1}^{n}\varepsilon_i \psi(h(\mathbf{x}_i))\right\|_\infty\right]$$

$$\leq \mu \mathbb{E}_{\boldsymbol{\varepsilon}}\left[\sup_{h \in \mathcal{H}_+} \left\|\frac{1}{n}\sum_{i=1}^{n}\varepsilon_i h(\mathbf{x}_i)\right\|_\infty\right] + \frac{1}{\sqrt{n}}\sup_{j \in [m]}\big|\psi_j(\mathit{0})\big|. \tag{71}$$

*Here, we define $\psi(\mathbf{x}) := (\psi(x_1), \psi(x_2), \cdots, \psi(x_m))^T$ for any $\mathbf{x} = (x_1, x_2, \cdots, x_m)^T \in \mathbb{R}^m$.*

Then, the following is a direct result of Lemma 17 by setting $\psi_j(\mathbf{x}) = \psi(x_j)$ for all $j \in [m], \mathbf{x} \in \mathbb{R}^m$ for some $\mu$-Lipschitz function $\psi : \mathbb{R} \to \mathbb{R}$.

**Corollary 18** *Let $\mathcal{H}$ be a set of functions mapping $\mathcal{X}$ to $\mathbb{R}^m$. Define*

$$\mathcal{H}_+ = \mathcal{H} \cup \big\{ - h : h \in \mathcal{H}\big\}. \tag{72}$$

*For any $\mu > 0$, let $\psi : \mathbb{R} \to \mathbb{R}$ such that $\big|\psi(x) - \psi(x')\big| \leq \mu|x - x'|, \ \forall(x, x') \in \mathbb{R} \times \mathbb{R}\}$ and $\psi - \psi(0)$ is odd. Then, it holds that*

$$\mathbb{E}_{\boldsymbol{\varepsilon}}\left[\sup_{h \in \mathcal{H}} \left\|\frac{1}{n}\sum_{i=1}^{n}\varepsilon_i \psi(h(\mathbf{x}_i))\right\|_\infty\right]$$

$$\leq \mu \mathbb{E}_{\boldsymbol{\varepsilon}}\left[\sup_{h \in \mathcal{H}_+} \left\|\frac{1}{n}\sum_{i=1}^{n}\varepsilon_i h(\mathbf{x}_i)\right\|_\infty\right] + \frac{1}{\sqrt{n}}\big|\psi(0)\big|. \tag{73}$$

*Here, we define $\psi(\mathbf{x}) := (\psi(x_1), \psi(x_2), \cdots, \psi(x_m))^T$ for any $\mathbf{x} = (x_1, x_2, \cdots, x_m)^T \in \mathbb{R}^m$.*

**Lemma 19** *Let $\mathcal{H}$ be a set of functions mapping $\mathcal{X}$ to $\mathbb{R}^m$. Define*

$$\mathcal{H}_+ = \mathcal{H} \cup \big\{ - h : h \in \mathcal{H}\big\} \cup \big\{|h| : h \in \mathcal{H}\big\}. \tag{74}$$

*For any $\mu > 0$, let $\psi : \mathbb{R} \to \mathbb{R}$ such that $\big|\psi(x) - \psi(x')\big| \leq \mu|x - x'|, \ \forall(x, x') \in \mathbb{R} \times \mathbb{R}\}$ and $\psi - \psi(0)$ is even. Then, it holds that*

$$\mathbb{E}_{\boldsymbol{\varepsilon}}\left[\sup_{h \in \mathcal{H}} \left\|\frac{1}{n}\sum_{i=1}^{n}\varepsilon_i \psi(h(\mathbf{x}_i))\right\|_\infty\right]$$

$$\leq 2\mu \mathbb{E}_{\boldsymbol{\varepsilon}}\left[\sup_{h \in \mathcal{H}_+} \left\|\frac{1}{n}\sum_{i=1}^{n}\varepsilon_i h(\mathbf{x}_i)\right\|_\infty\right] + \frac{1}{\sqrt{n}}\big|\psi(0)\big|. \tag{75}$$

*Here, we define $\psi(\mathbf{x}) := (\psi(x_1), \psi(x_2), \cdots, \psi(x_m))^T$ for any $\mathbf{x} = (x_1, x_2, \cdots, x_m)^T \in \mathbb{R}^m$.*

**Lemma 20** *Let $\mathcal{H}$ be a set of functions mapping $\mathcal{X}$ to $\mathbb{R}^m$. Define*

$$\mathcal{H}_+ = \mathcal{H} \cup \left\{ -h : h \in \mathcal{H} \right\} \cup \left\{ |h| : h \in \mathcal{H} \right\}. \tag{76}$$

*For any $\mu > 0$, let $\psi : \mathbb{R} \to \mathbb{R}$ such that $\left| \psi(x) - \psi(x') \right| \leq \mu |x - x'|, \ \forall (x, x') \in \mathbb{R} \times \mathbb{R} \}$. Then, it holds that*

$$\mathbb{E}_{\boldsymbol{\varepsilon}} \left[ \sup_{h \in \mathcal{H}} \left\| \frac{1}{n} \sum_{i=1}^{n} \varepsilon_i \psi(h(\mathbf{x}_i)) \right\|_{\infty} \right]$$

$$\leq 3\mu \mathbb{E}_{\boldsymbol{\varepsilon}} \left[ \sup_{h \in \mathcal{H}_+} \left\| \frac{1}{n} \sum_{i=1}^{n} \varepsilon_i h(\mathbf{x}_i) \right\|_{\infty} \right] + \frac{1}{\sqrt{n}} |\psi(0)|. \tag{77}$$

*Here, we define $\psi(\mathbf{x}) := (\psi(x_1), \psi(x_2), \cdots, \psi(x_m))^T$ for any $\mathbf{x} = (x_1, x_2, \cdots, x_m)^T \in \mathbb{R}^m$.*

These lemmas are proved in the next appendices.

## A.2 PROOF OF LEMMA 16

Observe that

$$\psi(x) = ReLU(\mathbf{x}) - \alpha ReLU(-x) \tag{78}$$

$$= \frac{x + |x|}{2} - \alpha \frac{-x + |x|}{2} \tag{79}$$

$$= \frac{1 + \alpha}{2} x + \frac{(1 - \alpha)}{2} |x|. \tag{80}$$

Then, for any $p \geq 1$ we have

$$\frac{1}{n} \mathbb{E}_{\boldsymbol{\varepsilon}} \left[ \sup_{h \in \mathcal{H}} \left\| \sum_{i=1}^{n} \varepsilon_i \psi(h(\mathbf{x}_i)) \right\|_{p} \right] \tag{81}$$

$$\leq \left( \frac{1 + \alpha}{2} \right) \frac{1}{n} \mathbb{E}_{\boldsymbol{\varepsilon}} \left[ \sup_{h \in \mathcal{H}} \left\| \sum_{i=1}^{n} \varepsilon_i h(\mathbf{x}_i) \right\|_{p} \right]$$

$$+ \left( \frac{1 - \alpha}{2} \right) \frac{1}{n} \mathbb{E}_{\boldsymbol{\varepsilon}} \left[ \sup_{h \in \mathcal{H}} \left\| \sum_{i=1}^{n} \varepsilon_i |h(\mathbf{x}_i)| \right\|_{p} \right] \tag{82}$$

$$\leq \frac{1}{n} \mathbb{E}_{\boldsymbol{\varepsilon}} \left[ \sup_{h \in \mathcal{H}_+} \left\| \sum_{i=1}^{n} \varepsilon_i h(\mathbf{x}_i) \right\|_{p} \right], \tag{83}$$

where (82) follows from Minkowski's inequality Royden & Fitzpatrick (2010), and (83) follows from the fact that $|h| \in \mathcal{H}_+$ if $h \in \mathcal{H}$.

## A.3 PROOF OF LEMMA 17

First, we have

$$\mathbb{E}_{\boldsymbol{\varepsilon}} \left[ \sup_{h \in \mathcal{H}} \left\| \frac{1}{n} \sum_{i=1}^{n} \varepsilon_i \psi(h(\mathbf{x}_i)) \right\|_{\infty} \right]$$

$$\leq \frac{1}{n} \left( \mathbb{E}_{\boldsymbol{\varepsilon}} \left[ \sup_{h \in \mathcal{H}} \left\| \sum_{i=1}^{n} \varepsilon_i \left( \psi(h(\mathbf{x}_i)) - \psi(\underline{0}) \right) \right\|_{\infty} \right] + \mathbb{E}_{\boldsymbol{\varepsilon}} \left[ \sup_{h \in \mathcal{H}} \left\| \sum_{i=1}^{n} \varepsilon_i \psi(\underline{0}) \right\|_{\infty} \right] \right) \tag{84}$$

$$\leq \frac{1}{n} \left( \mathbb{E}_{\boldsymbol{\varepsilon}} \left[ \sup_{h \in \mathcal{H}} \left\| \sum_{i=1}^{n} \varepsilon_i \left( \psi(h(\mathbf{x}_i)) - \psi(\underline{0}) \right) \right\|_{\infty} \right] + \mathbb{E}_{\boldsymbol{\varepsilon}} \left[ \left\| \sum_{i=1}^{n} \varepsilon_i \psi(\underline{0}) \right\|_{\infty} \right] \right) \tag{85}$$

$$\leq \frac{1}{n} \left( \mathbb{E}_{\boldsymbol{\varepsilon}} \left[ \sup_{h \in \mathcal{H}} \left\| \sum_{i=1}^{n} \varepsilon_i \left( \psi(h(\mathbf{x}_i)) - \psi(\underline{0}) \right) \right\|_{\infty} \right] + \sup_{j \in [m]} \sqrt{\mathbb{E}_{\boldsymbol{\varepsilon}} \left[ \left( \sum_{i=1}^{n} \varepsilon_i \psi_j(\underline{0}) \right)^2 \right]} \right) \tag{86}$$

$$\leq \frac{1}{n} \mathbb{E}_{\boldsymbol{\varepsilon}} \left[ \sup_{h \in \mathcal{H}} \left\| \sum_{i=1}^{n} \varepsilon_i \left( \psi(h(\mathbf{x}_i)) - \psi(\underline{0}) \right) \right\|_{\infty} \right] + \sup_{j \in [m]} |\psi_j(\underline{0})| \frac{1}{\sqrt{n}}, \tag{87}$$

where (84) follows from the triangular property of the $\infty$-norm Royden & Fitzpatrick (2010), and (86) follows from Cauchy-Schwarz inequality and the assumption that $\psi_j(\underline{0})$ does not depend on $j$.

Define $\tilde{\psi}(\mathbf{x}) := \psi(\mathbf{x}) - \psi(\underline{0})$ for all $\mathbf{x} \in \mathbb{R}^m$. Then, we have $\tilde{\psi}(\underline{0}) = \underline{0}$, and $\tilde{\psi}$ satisfies $|\tilde{\psi}_j(\mathbf{x}) - \tilde{\psi}_j(\mathbf{x}')| \leq \mu|x_j - x'_j|$ for all $\mathbf{x}, \mathbf{x}' \in \mathbb{R}^m, j \in [m]$. In addition, by our assumption, $\tilde{\psi}$ is odd.

Let

$$\Psi = \left\{ \tilde{\psi} : \mathbb{R}^m \to \mathbb{R}^m, \text{st. } \tilde{\psi}(-\mathbf{x}) = -\tilde{\psi}(\mathbf{x}), |\tilde{\psi}_j(\mathbf{x}) - \tilde{\psi}_j(\mathbf{y})| \leq \mu|x_j - y_j| \ \forall \mathbf{x}, \mathbf{y} \in \mathbb{R}^m, j \in [m] \right\}. \tag{88}$$

It follows that

$$\mathbb{E}_{\boldsymbol{\varepsilon}} \left[ \sup_{h \in \mathcal{H}} \left\| \frac{1}{n} \sum_{i=1}^{n} \varepsilon_i \tilde{\psi}\big(h(\mathbf{x}_i)\big) \right\|_{\infty} \right] \tag{89}$$

$$= \frac{1}{n} \mathbb{E}_{\boldsymbol{\varepsilon}} \left[ \sup_{j \in [m]} \sup_{h \in \mathcal{H}} \left| \sum_{i=1}^{n} \varepsilon_i \tilde{\psi}_j\big(h(\mathbf{x}_i)\big) \right| \right] \tag{90}$$

$$\leq \frac{1}{n} \mathbb{E}_{\boldsymbol{\varepsilon}} \left[ \sup_{s \in \{-1,+1\}^m} \sup_{j \in [m]} \sup_{h \in \mathcal{H}} s_j \left( \sum_{i=1}^{n} \varepsilon_i \tilde{\psi}_j\big(h(\mathbf{x}_i)\big) \right) \right] \tag{91}$$

$$= \frac{1}{n} \mathbb{E}_{\boldsymbol{\varepsilon}} \left[ \sup_{s \in \{-1,+1\}^m} \sup_{j \in [m]} \sup_{h \in \mathcal{H}} \sum_{i=1}^{n} \varepsilon_i s_j \tilde{\psi}_j\big(h(\mathbf{x}_i)\big) \right] \tag{92}$$

$$= \frac{1}{n} \mathbb{E}_{\boldsymbol{\varepsilon}} \left[ \sup_{s \in \{-1,+1\}^m} \sup_{j \in [m]} \sup_{h \in \mathcal{H}} \sum_{i=1}^{n} \varepsilon_i \tilde{\psi}_j^{(\mathbf{s})}\big(h(\mathbf{x}_i)\big) \right] \tag{93}$$

$$\leq \frac{1}{n} \mathbb{E}_{\boldsymbol{\varepsilon}} \left[ \sup_{\tilde{\psi} \in \Psi} \sup_{s \in \{-1,+1\}^m} \sup_{j \in [m]} \sup_{h \in \mathcal{H}} \sum_{i=1}^{n} \varepsilon_i \tilde{\psi}_j\big(h(\mathbf{x}_i)\big) \right] \tag{94}$$

$$\leq \frac{1}{n} \mathbb{E}_{\boldsymbol{\varepsilon}} \left[ \sup_{\tilde{\psi} \in \Psi} \sup_{j \in [m]} \sup_{h \in \mathcal{H}_+} \sum_{i=1}^{n} \varepsilon_i \tilde{\psi}_j\big(h(\mathbf{x}_i)\big) \right], \tag{95}$$

where (93) follows by defining $\tilde{\psi}^{(\mathbf{s})} = (s_1\tilde{\psi}_1, s_2\tilde{\psi}_2, \cdots, s_m\tilde{\psi}_m)$ for any $\mathbf{s} \in \{-1,+1\}^m$, (94) follows from the fact that $\tilde{\psi}^{(\mathbf{s})} \in \Psi$ for any fixed $\mathbf{s}$, and (95) follows from the definition of $\mathcal{H}_+$.

Now, we have

$$\mathbb{E}_{\boldsymbol{\varepsilon}} \left[ \sup_{\tilde{\psi} \in \Psi} \sup_{j \in [m]} \sup_{h \in \mathcal{H}_+} \sum_{i=1}^{n} \varepsilon_i \tilde{\psi}_j\big(h(\mathbf{x}_i)\big) \right]$$

$$= \mathbb{E}_{\varepsilon_1, \varepsilon_2, \cdots, \varepsilon_{n-1}} \left[ \mathbb{E}_{\varepsilon_n} \left[ \sup_{\tilde{\psi} \in \Psi} \sup_{j \in [m]} \sup_{h \in \mathcal{H}_+} u_{n-1}(h, j) + \varepsilon_n \tilde{\psi}_j\big(h(\mathbf{x}_n)\big) \right] \right], \tag{96}$$

where

$$u_{n-1}(h, j) := \sum_{i=1}^{n-1} \varepsilon_i \tilde{\psi}_j\big(h(\mathbf{x}_i)\big). \tag{97}$$

Since $\varepsilon_n$ is uniformly distributed over $\{-1, 1\}$, we have

$$\mathbb{E}_{\varepsilon_n} \left[ \sup_{\tilde{\psi} \in \Psi} \sup_{j \in [m]} \sup_{h \in \mathcal{H}_+} u_{n-1}(h, j) + \varepsilon_n \tilde{\psi}_j\big(h(\mathbf{x}_n)\big) \right]$$

$$= \frac{1}{2} \left( \sup_{\tilde{\psi} \in \Psi} \sup_{j \in [m]} \sup_{h \in \mathcal{H}_+} u_{n-1}(h, j) + \tilde{\psi}_j(h(\mathbf{x}_n)) \right)$$

$$+ \frac{1}{2} \left( \sup_{\tilde{\psi} \in \Psi} \sup_{j \in [m]} \sup_{h \in \mathcal{H}_+} u_{n-1}(h, j) - \tilde{\psi}_j(h(\mathbf{x}_n)) \right). \tag{98}$$

Hence, we have

$$\mathbb{E}_{\boldsymbol{\varepsilon}}\left[\sup_{\tilde{\psi}\in\Psi}\sup_{j\in[m]}\sup_{h\in\mathcal{H}_+}\sum_{i=1}^{n}\varepsilon_i\tilde{\psi}_j\big(h(\mathbf{x}_i)\big)\right]$$

$$=\frac{1}{2}\mathbb{E}_{\varepsilon_1,\varepsilon_2,\cdots,\varepsilon_{n-1}}\left[\sup_{\tilde{\psi}\in\Psi}\sup_{j\in[m]}\sup_{h\in\mathcal{H}_+}u_{n-1}(h,j)+\tilde{\psi}_j(h(\mathbf{x}_n))\right]$$

$$+\frac{1}{2}\mathbb{E}_{\varepsilon_1,\varepsilon_2,\cdots,\varepsilon_{n-1}}\left[\sup_{\tilde{\psi}\in\Psi}\sup_{j\in[m]}\sup_{h\in\mathcal{H}_+}u_{n-1}(h,j)-\tilde{\psi}_j(h(\mathbf{x}_n))\right] \tag{99}$$

$$=\frac{1}{2}\mathbb{E}_{\varepsilon_1,\varepsilon_2,\cdots,\varepsilon_{n-1}}\left[\sup_{\tilde{\psi}\in\Psi}\sup_{j\in[m]}\sup_{h\in\mathcal{H}_+}u_{n-1}(h,j)+\tilde{\psi}_j(h(\mathbf{x}_n))\right]$$

$$+\frac{1}{2}\mathbb{E}_{\varepsilon_1,\varepsilon_2,\cdots,\varepsilon_{n-1}}\left[\sup_{\tilde{\psi}\in\Psi}\sup_{j\in[m]}\sup_{h\in\mathcal{H}_+}-u_{n-1}(h,j)-\tilde{\psi}_j(h(\mathbf{x}_n))\right] \tag{100}$$

$$=\mathbb{E}_{\varepsilon_1,\varepsilon_2,\cdots,\varepsilon_{n-1}}\left[\frac{1}{2}\left(\sup_{\tilde{\psi}\in\Psi}\sup_{j\in[m]}\sup_{h\in\mathcal{H}_+}u_{n-1}(h,j)+\tilde{\psi}_j(h(\mathbf{x}_n))\right)\right.$$

$$\left.+\frac{1}{2}\left(\sup_{\tilde{\psi}\in\Psi}\sup_{j\in[m]}\sup_{h\in\mathcal{H}_+}-u_{n-1}(h,j)-\tilde{\psi}_j(h(\mathbf{x}_n))\right)\right], \tag{101}$$

where (100) follows from the fact that $(-\varepsilon_1,-\varepsilon_2,\cdots,-\varepsilon_{n-1})$ is a tuple of independent Rademacher random variables which has the same distribution as $(\varepsilon_1,\varepsilon_2,\cdots,\varepsilon_{n-1})$.

Now, given any $j\in[m]$ and $\tilde{\psi}\in\Psi$ we have

$$\sup_{h\in\mathcal{H}_+}u_{n-1}(h,j)+\tilde{\psi}_j(h(\mathbf{x}_n))$$

$$=\sup_{h\in\mathcal{H}_+}u_{n-1}(-h,j)+\tilde{\psi}_j(-h(\mathbf{x}_n)) \tag{102}$$

$$=\sup_{h\in\mathcal{H}_+}-u_{n-1}(h,j)-\tilde{\psi}_j(h(\mathbf{x}_n)), \tag{103}$$

where (102) follows from the assumption that $h\in\mathcal{H}_+$ if and only if $-h\in\mathcal{H}_+$, and (103) follows from the assumption that $\tilde{\psi}$ is odd for any $\tilde{\psi}\in\Psi$.

Hence, for any arbitrarily small $\delta>0$ there exists $j_0\in[m],\tilde{\psi}_0\in\Psi$ and $h_1,h_2\in\mathcal{H}$ such that

$$\sup_{\tilde{\psi}\in\Psi}\sup_{j\in[m]}\sup_{h\in\mathcal{H}_+}u_{n-1}(h,j)+\tilde{\psi}_j(h(\mathbf{x}_n))\leq u_{n-1}(h_1,j_0)+\tilde{\psi}_{0,j_0}(h_1(\mathbf{x}_n))+\delta, \tag{104}$$

and

$$\sup_{\tilde{\psi}\in\Psi}\sup_{j\in[m]}\sup_{h\in\mathcal{H}_+}-u_{n-1}(h,j)-\tilde{\psi}([h(\mathbf{x}_n)]_j)\leq -u_{n-1}(h_2,j_0)-\tilde{\psi}_{0,j_0}(h_2(\mathbf{x}_n))+\delta. \tag{105}$$

It follows that

$$
\frac{1}{2}\Bigg( \sup_{\tilde\psi\in\Psi} \sup_{j\in[m]} \sup_{h\in\mathcal{H}_+} u_{n-1}(h,j) + \tilde\psi_j(h(\mathbf{x}_n)) \Bigg)
$$

$$
+ \frac{1}{2}\Bigg( \sup_{\tilde\psi\in\Psi} \sup_{j\in[m]} \sup_{h\in\mathcal{H}_+} -u_{n-1}(h,j) - \tilde\psi_j(h(\mathbf{x}_n)) \Bigg)
$$

$$
\leq \frac{1}{2}\Big( u_{n-1}(h_1,j_0) + \tilde\psi_{0,j_0}(h_1(\mathbf{x}_n)) \Big)
$$

$$
+ \frac{1}{2}\Big( -u_{n-1}(h_2,j_0) - \tilde\psi_{0,j_0}(h_2(\mathbf{x}_n)) \Big) + \delta \tag{106}
$$

$$
= \frac{1}{2}\big( u_{n-1}(h_1,j_0) - u_{n-1}(h_2,j_0) \big)
$$

$$
+ \frac{1}{2}\big( \tilde\psi_{0,j_0}(h_1(\mathbf{x}_n)) - \tilde\psi_{0,j_0}(h_2(\mathbf{x}_n)) \big) + \delta \tag{107}
$$

$$
\leq \frac{1}{2}\big( u_{n-1}(h_1,j_0) - u_{n-1}(h_2,j_0) \big) + \frac{\mu}{2}\big| [h_1(\mathbf{x}_n)]_{j_0} - [h_2(\mathbf{x}_n)]_{j_0} \big| \tag{108}
$$

$$
= \frac{1}{2}\big( u_{n-1}(h_1,j_0) - u_{n-1}(h_2,j_0) \big) + \frac{\mu}{2} s_{12,n}\big( [h_1(\mathbf{x}_n)]_{j_0} - [h_2(\mathbf{x}_n)]_{j_0} \big) \tag{109}
$$

$$
= \frac{1}{2}\big( u_{n-1}(h_1,j_0) + \mu s_{12,n}[h_1(\mathbf{x}_n)]_{j_0} \big) + \frac{1}{2}\big( -u_{n-1}(h_2,j_0) - \mu s_{12,n}[h_2(\mathbf{x}_n)]_{j_0} \big) \tag{110}
$$

$$
\leq \sup_{s_{12}\in\{-1,+1\}} \frac{1}{2}\big( u_{n-1}(h_1,j_0) + \mu s_{12}[h_1(\mathbf{x}_n)]_{j_0} \big) + \frac{1}{2}\big( -u_{n-1}(h_2,j_0) - \mu s_{12}[h_2(\mathbf{x}_n)]_{j_0} \big)
$$
$$
\tag{111}
$$

$$
\leq \sup_{s_{12}\in\{-1,+1\}} \frac{1}{2} \sup_{\tilde\psi\in\Psi} \sup_{j\in[m]} \sup_{h\in\mathcal{H}_+} u_{n-1}(h,j) + \mu s_{12}[h(\mathbf{x}_n)]_j
$$

$$
+ \frac{1}{2} \sup_{\tilde\psi\in\Psi} \sup_{j\in[m]} \sup_{h\in\mathcal{H}_+} -u_{n-1}(h,j) - \mu s_{12}[h(\mathbf{x}_n)]_j \tag{112}
$$

$$
\leq \sup_{s_{12}\in\{-1,+1\}} \frac{1}{2} \sup_{\tilde\psi\in\Psi} \sup_{j\in[m]} \sup_{h\in\mathcal{H}_+} u_{n-1}(h,j) + \mu s_{12}[h(\mathbf{x}_n)]_j
$$

$$
+ \frac{1}{2} \sup_{\tilde\psi\in\Psi} \sup_{j\in[m]} \sup_{h\in\mathcal{H}_+} u_{n-1}(h,j) - \mu s_{12}[h(\mathbf{x}_n)]_j, \tag{113}
$$

where $s_{12,n} := \mathrm{sgn}\big([h_1(\mathbf{x}_n)]_{j_0} - [h_2(\mathbf{x}_n)]_{j_0}\big)$ in (109), and (113) follows from the fact that $-\tilde\psi\in\Psi$ if $\tilde\psi\in\Psi$.

From (113) we obtain

$$
\frac{1}{2}\Bigg( \sup_{\tilde\psi\in\Psi} \sup_{j\in[m]} \sup_{h\in\mathcal{H}_+} u_{n-1}(h,j) + \tilde\psi_j(h(\mathbf{x}_n)) \Bigg)
$$

$$
+ \frac{1}{2}\Bigg( \sup_{\tilde\psi\in\Psi} \sup_{j\in[m]} \sup_{h\in\mathcal{H}_+} -u_{n-1}(h,j) - \tilde\psi_j(h(\mathbf{x}_n)) \Bigg) \tag{114}
$$

$$
\leq \sup_{s_{12}\in\{-1,+1\}} \mathbb{E}_{\tilde\varepsilon_n}\Bigg[ \sup_{\tilde\psi\in\Psi} \sup_{j\in[m]} \sup_{h\in\mathcal{H}_+} u_{n-1}(h,j) + \mu\tilde\varepsilon_n s_{12}[h(\mathbf{x}_n)]_j \Bigg] \tag{115}
$$

for some Rademacher random variable $\tilde\varepsilon_n$ which is independent of $(\varepsilon_1,\varepsilon_2,\cdots,\varepsilon_{n-1})$.

Since $\tilde{\varepsilon}_n s_{12} \sim \tilde{\varepsilon}_n$ for any fixed $s_{12} \in \{-1, +1\}$, from (115) we have

$$\frac{1}{2}\left( \sup_{\tilde{\psi} \in \Psi} \sup_{j \in [m]} \sup_{h \in \mathcal{H}_+} u_{n-1}(h, j) + \tilde{\psi}_j(h(\mathbf{x}_n)) \right)$$

$$+ \frac{1}{2}\left( \sup_{\tilde{\psi} \in \Psi} \sup_{j \in [m]} \sup_{h \in \mathcal{H}_+} -u_{n-1}(h, j) - \tilde{\psi}_j(h(\mathbf{x}_n)) \right) \tag{116}$$

$$\leq \mathbb{E}_{\tilde{\varepsilon}_n}\left[ \sup_{\tilde{\psi} \in \Psi} \sup_{j \in [m]} \sup_{h \in \mathcal{H}_+} u_{n-1}(h, j) + \mu\tilde{\varepsilon}_n[h(\mathbf{x}_n)]_j \right]. \tag{117}$$

From (101) and (117) we obtain

$$\mathbb{E}_{\boldsymbol{\varepsilon}}\left[ \sup_{\tilde{\psi} \in \Psi} \sup_{j \in [m]} \sup_{h \in \mathcal{H}_+} \sum_{i=1}^n \varepsilon_i \tilde{\psi}_j\big(h(\mathbf{x}_i)\big) \right]$$

$$\leq \mathbb{E}_{\varepsilon_1, \varepsilon_2, \cdots, \varepsilon_{n-1}}\left[ \mathbb{E}_{\tilde{\varepsilon}_n}\left[ \sup_{\tilde{\psi} \in \Psi} \sup_{j \in [m]} \sup_{h \in \mathcal{H}_+} u_{n-1}(h, j) + \mu\tilde{\varepsilon}_n[h(\mathbf{x}_n)]_j \right] \right] \tag{118}$$

$$= \mathbb{E}_{\tilde{\varepsilon}_n}\left[ \mathbb{E}_{\varepsilon_1, \varepsilon_2, \cdots, \varepsilon_{n-1}}\left[ \sup_{\tilde{\psi} \in \Psi} \sup_{j \in [m]} \sup_{h \in \mathcal{H}_+} u_{n-1}(h, j) + \mu\tilde{\varepsilon}_n[h(\mathbf{x}_n)]_j \right] \right]. \tag{119}$$

By continuing this process (peeling) for $n - 1$ more times, we have

$$\mathbb{E}_{\boldsymbol{\varepsilon}}\left[ \sup_{\tilde{\psi} \in \Psi} \sup_{j \in [m]} \sup_{h \in \mathcal{H}_+} u_{n-1}(h, j) + \tilde{\varepsilon}_n \mu[h(\mathbf{x}_n)]_j \right]$$

$$\leq \mu \mathbb{E}_{\tilde{\varepsilon}_1, \tilde{\varepsilon}_2, \cdots, \tilde{\varepsilon}_n}\left[ \sup_{j \in [m]} \sup_{h \in \mathcal{H}_+} \sum_{i=1}^n \tilde{\varepsilon}_i[h(\mathbf{x}_n)]_j \right] \tag{120}$$

$$= \mu \mathbb{E}_{\boldsymbol{\varepsilon}}\left[ \sup_{j \in [m]} \sup_{h \in \mathcal{H}_+} \sum_{i=1}^n \varepsilon_i[h(\mathbf{x}_n)]_j \right] \tag{121}$$

$$\leq \mu \mathbb{E}_{\boldsymbol{\varepsilon}}\left[ \sup_{j \in [m]} \sup_{h \in \mathcal{H}_+} \left| \sum_{i=1}^n \varepsilon_i[h(\mathbf{x}_n)]_j \right| \right] \tag{122}$$

$$= \mu \mathbb{E}_{\boldsymbol{\varepsilon}}\left[ \sup_{h \in \mathcal{H}_+} \left\| \sum_{i=1}^n \varepsilon_i h(\mathbf{x}_n) \right\|_\infty \right]. \tag{123}$$

From (87) and (123), we obtain

$$\mathbb{E}_{\boldsymbol{\varepsilon}}\left[ \sup_{h \in \mathcal{H}} \left\| \frac{1}{n} \sum_{i=1}^n \varepsilon_i \psi(h(\mathbf{x}_i)) \right\|_\infty \right]$$

$$\leq \mu \mathbb{E}_{\boldsymbol{\varepsilon}}\left[ \sup_{h \in \mathcal{H}_+} \left\| \frac{1}{n} \sum_{i=1}^n \varepsilon_i h(\mathbf{x}_n) \right\|_\infty \right] + \sup_{j \in [m]} \left| \psi_j(0) \right| \frac{1}{\sqrt{n}}. \tag{124}$$

This concludes our proof of Lemma 17.

### A.4 PROOF OF LEMMA 19

Since $\psi(x)$ is even, it holds that

$$\mathbb{E}\left[ \sup_{h \in \mathcal{H}} \frac{1}{n} \left\| \sum_{i=1}^n \varepsilon_i \psi(h(\mathbf{x}_i)) \right\|_\infty \right] = \mathbb{E}\left[ \sup_{h \in \mathcal{H}} \frac{1}{n} \left\| \sum_{i=1}^n \varepsilon_i \psi\big(|h(\mathbf{x}_i)|\big) \right\|_\infty \right], \tag{125}$$

Define

$$\tilde{\psi}(x) := \psi\big(x\mathbf{1}\{x > 0\}\big) - \psi\big(-x\mathbf{1}\{x < 0\}\big) \qquad \forall x \in \mathbb{R}. \tag{126}$$

Then, it is easy to see that $\tilde{\psi}$ is an odd function.

On the other hand, we also have

$$\tilde{\psi}(|x|) = \psi(|x|), \qquad \forall x \in \mathbb{R}, \tag{127}$$

so

$$\mathbb{E}\left[\sup_{h\in\mathcal{H}} \frac{1}{n}\left\|\sum_{i=1}^{n} \varepsilon_i \psi(|h(\mathbf{x}_i)|)\right\|_{\infty}\right] = \mathbb{E}\left[\sup_{h\in\mathcal{H}} \frac{1}{n}\left\|\sum_{i=1}^{n} \varepsilon_i \tilde{\psi}(|h(\mathbf{x}_i)|)\right\|_{\infty}\right]. \tag{128}$$

Furthermore, for all $x, y \in \mathbb{R}$ we have

$$\left|\tilde{\psi}(x) - \tilde{\psi}(y)\right|$$
$$\leq \left|\psi(x\mathbf{1}\{x > 0\}) - \psi(y\mathbf{1}\{y > 0\})\right| + \left|\psi(x\mathbf{1}\{x < 0\}) - \psi(y\mathbf{1}\{y < 0\})\right| \tag{129}$$
$$\leq \mu\left|x\mathbf{1}\{x > 0\} - y\mathbf{1}\{y > 0\}\right| + \mu\left|x\mathbf{1}\{x < 0\} - y\mathbf{1}\{y < 0\}\right| \tag{130}$$

Now, observe that

$$\left|x\mathbf{1}\{x > 0\} - y\mathbf{1}\{y > 0\}\right|$$
$$= \left|\frac{x + |x|}{2} - \frac{y + |y|}{2}\right| \tag{131}$$
$$\leq \frac{1}{2}|x - y| + \frac{1}{2}\sum_{i=1}^{L} \left||x| - |y|\right| \tag{132}$$
$$\leq |x - y| \tag{133}$$

Similarly, we also have

$$\left|x\mathbf{1}\{x < 0\} - y\mathbf{1}\{y < 0\}\right| \leq |x - y|. \tag{134}$$

From (130), (133), and (134) we obtain

$$\left|\tilde{\psi}(x) - \tilde{\psi}(y)\right| \leq 2\mu|x - y|, \qquad \forall x, y \in \mathbb{R}. \tag{135}$$

Hence, by Lemma 18 we have

$$\mathbb{E}\left[\sup_{h\in\mathcal{H}} \frac{1}{n}\left\|\sum_{i=1}^{n} \varepsilon_i \tilde{\psi}(|h(\mathbf{x}_i)|)\right\|_{\infty}\right]$$
$$\leq 2\mu\mathbb{E}\left[\sup_{h\in\mathcal{H}_+} \frac{1}{n}\left\|\sum_{i=1}^{n} \varepsilon_i |h(\mathbf{x}_i)|\right\|_{\infty}\right] \tag{136}$$
$$\leq 2\mu\mathbb{E}\left[\sup_{h\in\mathcal{H}_+} \frac{1}{n}\left\|\sum_{i=1}^{n} \varepsilon_i h(\mathbf{x}_i)\right\|_{\infty}\right], \tag{137}$$

where (137) follows by using the fact that $|h| \in \mathcal{H}$ if $h \in \mathcal{H}_+$.

Hence, finally we have

$$\mathbb{E}\left[\sup_{h\in\mathcal{H}} \frac{1}{n}\left\|\sum_{i=1}^{n} \varepsilon_i \psi(h(\mathbf{x}_i))\right\|_{\infty}\right] \leq 2\mu\mathbb{E}\left[\sup_{h\in\mathcal{H}_+} \frac{1}{n}\left\|\sum_{i=1}^{n} \varepsilon_i h(\mathbf{x}_i)\right\|_{\infty}\right]. \tag{138}$$

## A.5 PROOF OF LEMMA 20

For any general function $\psi$, we can represent as

$$\psi(x) = \frac{\psi(x) + \psi(-x)}{2} + \frac{\psi(x) - \psi(-x)}{2}, \qquad \forall \mathbf{x} \in \mathbb{R}. \tag{139}$$

It is easy to see that $\frac{\psi(x)+\psi(-x)}{2}$ is an even function with $\mu$-Lipschitz. Besides, $\frac{\psi(x)-\psi(-x)}{2}$ is an odd function with $\mu$-Lischitz. Hence, by using triangle inequality, Lemma 17 and Lemma 19, we have

$$\mathbb{E}\left[\sup_{h\in\mathcal{H}} \frac{1}{n}\left\|\sum_{i=1}^{n} \varepsilon_i \psi(h(\mathbf{x}_i))\right\|_{\infty}\right] \leq (2\mu + \mu)\mathbb{E}\left[\sup_{h\in\mathcal{H}_+} \frac{1}{n}\left\|\sum_{i=1}^{n} \varepsilon_i h(\mathbf{x}_i)\right\|_{\infty}\right]. \tag{140}$$

## A.6 PROOF OF THEOREM 4

For any $\mathbf{W} \in \mathcal{V}$, observe that

$$\left\|\frac{1}{n}\sum_{i=1}^{n}\varepsilon_i\mathbf{W}f(\mathbf{x}_i)\right\|_{\infty} = \left\|\mathbf{W}\left(\frac{1}{n}\sum_{i=1}^{n}\varepsilon_if(\mathbf{x}_i)\right)\right\|_{\infty} \tag{141}$$

$$\leq \left\|\mathbf{W}\right\|_{\infty}\left\|\frac{1}{n}\sum_{i=1}^{n}\varepsilon_if(\mathbf{x}_i)\right\|_{\infty} \tag{142}$$

$$\leq \nu\left\|\frac{1}{n}\sum_{i=1}^{n}\varepsilon_if(\mathbf{x}_i)\right\|_{\infty}. \tag{143}$$

Hence, (13) is a direct application of this fact.

This concludes our proof of Theorem 4.

## A.7 PROOF OF LEMMA 6

Let

$$\mathbf{1}_{\tau_l^2} = \underbrace{\begin{bmatrix} 1 & 1 & \cdots & 1 \end{bmatrix}}_{\tau_l^2}, \tag{144}$$

$$0_{\tau_l^2} = \underbrace{\begin{bmatrix} 0 & 0 & \cdots & 0 \end{bmatrix}}_{\tau_l^2}, \tag{145}$$

and

$$\mathbf{A} = \frac{1}{\tau_l^2}\begin{bmatrix} \mathbf{1}_{\tau_l^2} & 0_{\tau_l^2} & 0_{\tau_l^2} & \cdots & 0_{\tau_l^2} & 0_{\tau_l^2} \\ 0_{\tau_l^2} & \mathbf{1}_{\tau_l^2} & 0_{\tau_l^2} & \cdots & 0_{\tau_l^2} & 0_{\tau_l^2} \\ \vdots & \vdots & \vdots & \ddots & \vdots & \vdots \\ 0_{\tau_l^2} & 0_{\tau_l^2} & 0_{\tau_l^2} & \cdots & 0_{\tau_l^2} & \mathbf{1}_{\tau_l^2} \end{bmatrix} \in \mathbb{R}^{\lceil(d-r_l+1)^2/\tau_l^2\rceil\tau_l^2 \times \lceil(d-r_l+1)^2/\tau_l^2\rceil\tau_l^2}. \tag{146}$$

Then, for all $\mathbf{x} \in \mathbb{R}^{d \times d \times C}$ and $l \in [Q], c \in [C]$, we have

$$\psi_{l,c}(\mathbf{x}) = \sigma_{\text{avg}} \circ \sigma_{l,c}(\mathbf{x}), \tag{147}$$

where

$$\sigma_{\text{avg}}(\mathbf{x}) = \mathbf{A}\mathbf{x}, \qquad \forall \mathbf{x} \in \mathbb{R}^{\lceil(d-r_l+1)^2/\tau_l^2\rceil\tau_l^2}. \tag{148}$$

Now, for all $\mathbf{x}, \mathbf{y} \in \mathbb{R}^{\lceil(d-r_l+1)^2/\tau_l^2\rceil\tau_l^2}$ we have

$$\left\|\sigma_{\text{avg}}(\mathbf{x}) - \sigma_{\text{avg}}(\mathbf{y})\right\|_{\infty}$$

$$\leq \frac{1}{\tau_l^2}\max_{j\in[\lceil(d-r_l+1)^2/\tau_l^2\rceil]}\sum_{k=(j-1)\tau_l^2+1}^{j\tau_l^2}\left|x_k - y_k\right| \tag{149}$$

$$\leq \left\|\mathbf{x} - \mathbf{y}\right\|_{\infty}. \tag{150}$$

Hence, we have

$$\left\|\mathbf{A}\right\|_{\infty} \leq 1. \tag{151}$$

Hence, by Lemma 4 we have

$$\mathbb{E}\left[\sup_{c\in[C]}\sup_{l\in[Q]}\sup_{\psi_{l,c}\in\Psi}\sup_{f\in\mathcal{F}}\left\|\frac{1}{n}\sum_{i=1}^{n}\varepsilon_i\psi_{l,c}\circ f(\mathbf{x}_i)\right\|_{\infty}\right]$$

$$= \mathbb{E}\left[\sup_{c\in[C]}\sup_{l\in[Q]}\sup_{\sigma_{\text{avg}}}\sup_{\sigma_{l,c}}\sup_{f\in\mathcal{F}}\left\|\frac{1}{n}\sum_{i=1}^{n}\varepsilon_i\sigma_{\text{avg}}\circ\sigma_{l,c}\circ f(\mathbf{x}_i)\right\|_{\infty}\right] \tag{152}$$

$$\leq \mathbb{E}\left[\sup_{c\in[C]}\sup_{l\in[Q]}\sup_{\sigma_{l,c}}\sup_{f\in\mathcal{F}}\left\|\frac{1}{n}\sum_{i=1}^{n}\varepsilon_i\sigma_{l,c}\circ f(\mathbf{x}_i)\right\|_{\infty}\right]. \tag{153}$$

In addition, for all $\mathbf{x} \in \mathbb{R}^{d \times d \times C}$,

$$\sigma_{l,c}(\mathbf{x}) = \{\hat{x}_c(a,b)\}_{a,b=1}^{d-r_l+1}, \tag{154}$$

$$\hat{x}_c(a,b) = \sigma\left(\sum_{u=0}^{r_l-1}\sum_{v=0}^{r_l-1} x(a+u,b+v,c)W_{l,c}(u+1,v+1)\right). \tag{155}$$

Hence, we have

$$\left\|\sigma_{l,c}(\mathbf{x}) - \sigma_{l,c}(\mathbf{y})\right\|_\infty$$

$$\leq \mu \max_{a\in[d-r_l+1]} \max_{b\in[d-r_l+1]} \sum_{u=0}^{r_l-1}\sum_{v=0}^{r_l-1} \Big| W_{l,c}(u+1,v+1)x(a+u,b+v,c)$$

$$- W_{l,c}(u+1,v+1)y(a+u,b+v,c)\Big| \tag{156}$$

$$\leq \mu \sum_{u=0}^{r_l-1}\sum_{v=0}^{r_l-1} \big|W_{l,c}(u+1,v+1)\big|\|\mathbf{x}-\mathbf{y}\|_\infty. \tag{157}$$

Since the convolution is linear, it is also easy to see that $\sigma_{l,c}$ is the composition of a linear map and a point-wise activation map. Hence, by Lemma 4 and Theorem 2 we have

$$\mathbb{E}\left[\sup_{c\in[C]}\sup_{l\in[Q]}\sup_{\sigma_{l,c}}\sup_{f\in\mathcal{F}}\left\|\frac{1}{n}\sum_{i=1}^n \varepsilon_i\sigma_{l,c}\circ f(\mathbf{x}_i)\right\|_\infty\right]$$

$$\leq \left[\gamma(\mu)\sup_{c\in[C]}\sup_{l\in[Q]}\left(\sum_{u=0}^{r_l-1}\sum_{v=0}^{r_l-1}\big|W_{l,c}(u+1,v+1)\big|\right)\right]\mathbb{E}\left[\sup_{f\in\mathcal{F}_+}\left\|\frac{1}{n}\sum_{i=1}^n \varepsilon_i f(\mathbf{x}_i)\right\|_\infty\right] + \frac{|\sigma(0)|}{\sqrt{n}}. \tag{158}$$

Finally, from (153) and (158) we obtain

$$\mathbb{E}\left[\sup_{c\in[C]}\sup_{l\in[Q]}\sup_{\psi_{l,c}\in\Psi}\sup_{f\in\mathcal{F}}\left\|\frac{1}{n}\sum_{i=1}^n \varepsilon_i\psi_{l,c}\circ f(\mathbf{x}_i)\right\|_\infty\right]$$

$$\leq \left[\gamma(\mu)\sup_{c\in[C]}\sup_{l\in[Q]}\left(\sum_{u=0}^{r_l-1}\sum_{v=0}^{r_l-1}\big|W_{l,c}(u+1,v+1)\big|\right)\right]\mathbb{E}\left[\sup_{f\in\mathcal{F}_+}\left\|\frac{1}{n}\sum_{i=1}^n \varepsilon_i f(\mathbf{x}_i)\right\|_\infty\right] + \frac{|\sigma(0)|}{\sqrt{n}}. \tag{159}$$

## A.8 PROOF OF LEMMA 7

This is a direct result of Lemma 17, where $\tilde{\psi}_j(\mathbf{x}) = x_j$ or $0$ at each fixed $j$. Hence, we have

$$\big|\tilde{\psi}_j(\mathbf{x}) - \tilde{\psi}_j(\mathbf{y})\big| \leq |x_j - y_j| \tag{160}$$

for all vectors $\mathbf{x}$ and $\mathbf{y}$.

## A.9 PROOF OF LEMMA 8

This is a direct result of Theorem 2 and Lemma 4.

## A.10 PROOF OF LEMMA 13

For $M > 2$, (52) is a result of (Koltchinskii & Panchenko, 2002, Proof of Theorem 11). Now, we prove (52) for $M = 2$. Observe that

$$\mathbb{E}_{\boldsymbol{\varepsilon}}\left[\sup_{f\in\mathcal{F}}\left|\frac{1}{n}\sum_{i=1}^n \varepsilon_i m_f(\mathbf{x}_i, y_i)\right|\right]$$

$$= \mathbb{E}_{\boldsymbol{\varepsilon}}\left[\sup_{f\in\mathcal{F}}\left|\frac{1}{n}\sum_{i=1}^n \varepsilon_i\left([f(\mathbf{x}_i)]_{y_i} - \sup_{y'\neq y_i}[f(\mathbf{x}_i)]_{y'}\right)\right|\right] \tag{161}$$

$$\leq \mathbb{E}_{\boldsymbol{\varepsilon}}\left[\sup_{f\in\mathcal{F}}\left|\frac{1}{n}\sum_{i=1}^n \varepsilon_i[f(\mathbf{x}_i)]_{y_i}\right|\right] + \mathbb{E}_{\boldsymbol{\varepsilon}}\left[\sup_{f\in\mathcal{F}}\left|\frac{1}{n}\sum_{i=1}^n \varepsilon_i\sup_{y'\neq y_i}[f(\mathbf{x}_i)]_{y'}\right|\right]. \tag{162}$$

Now, we have

$$\mathbb{E}_{\boldsymbol{\varepsilon}}\left[\sup_{f\in\mathcal{F}}\left|\frac{1}{n}\sum_{i=1}^{n}\varepsilon_i[f(\mathbf{x}_i)]_{y_i}\right|\right]$$

$$= \mathbb{E}_{\boldsymbol{\varepsilon}}\left[\sup_{f\in\mathcal{F}}\left|\frac{1}{n}\sum_{i=1}^{n}\varepsilon_i[f(\mathbf{x}_i)]_{y_i}\sum_{y=1}^{M}\mathbf{1}_{\{y_i=y\}}\right|\right] \tag{163}$$

$$= \mathbb{E}_{\boldsymbol{\varepsilon}}\left[\sup_{f\in\mathcal{F}}\left|\frac{1}{n}\sum_{y=1}^{M}\sum_{i=1}^{n}\varepsilon_i[f(\mathbf{x}_i)]_{y}\mathbf{1}_{\{y_i=y\}}\right|\right] \tag{164}$$

$$\leq \sum_{y=1}^{M}\mathbb{E}_{\boldsymbol{\varepsilon}}\left[\sup_{f\in\mathcal{F}}\left|\frac{1}{n}\sum_{i=1}^{n}\varepsilon_i[f(\mathbf{x}_i)]_{y}\mathbf{1}_{\{y_i=y\}}\right|\right] \tag{165}$$

$$\leq \frac{1}{2}\sum_{y=1}^{M}\mathbb{E}_{\boldsymbol{\varepsilon}}\left[\sup_{f\in\mathcal{F}}\left|\frac{1}{n}\sum_{i=1}^{n}\varepsilon_i[f(\mathbf{x}_i)]_{y}(2\mathbf{1}_{\{y_i=y\}}-1)\right|\right]$$

$$+ \frac{1}{2}\sum_{y=1}^{M}\mathbb{E}_{\boldsymbol{\varepsilon}}\left[\sup_{f\in\mathcal{F}}\left|\frac{1}{n}\sum_{i=1}^{n}\varepsilon_i[f(\mathbf{x}_i)]_{y}\right|\right] \tag{166}$$

$$= \frac{1}{2}\sum_{y=1}^{M}\mathbb{E}_{\boldsymbol{\varepsilon}}\left[\sup_{f\in\mathcal{F}}\left|\frac{1}{n}\sum_{i=1}^{n}\varepsilon_i[f(\mathbf{x}_i)]_{y}\right|\right]$$

$$+ \frac{1}{2}\sum_{y=1}^{M}\mathbb{E}_{\boldsymbol{\varepsilon}}\left[\sup_{f\in\mathcal{F}}\left|\frac{1}{n}\sum_{i=1}^{n}\varepsilon_i[f(\mathbf{x}_i)]_{y}\right|\right] \tag{167}$$

$$= \sum_{y=1}^{M}\mathbb{E}_{\boldsymbol{\varepsilon}}\left[\sup_{f\in\mathcal{F}}\left|\frac{1}{n}\sum_{i=1}^{n}\varepsilon_i[f(\mathbf{x}_i)]_{y}\right|\right] \tag{168}$$

$$\leq \sum_{y=1}^{M}\mathbb{E}_{\boldsymbol{\varepsilon}}\left[\sup_{f\in\mathcal{F}}\left\|\frac{1}{n}\sum_{i=1}^{n}\varepsilon_i f(\mathbf{x}_i)\right\|_{\infty}\right] \tag{169}$$

$$= M\mathbb{E}_{\boldsymbol{\varepsilon}}\left[\sup_{f\in\mathcal{F}}\left\|\frac{1}{n}\sum_{i=1}^{n}\varepsilon_i f(\mathbf{x}_i)\right\|_{\infty}\right], \tag{170}$$

where (167) follows from the fact that $(2\mathbf{1}_{\{y_1=y\}}-1)\varepsilon_1, (2\mathbf{1}_{\{y_2=y\}}-1)\varepsilon_2, \cdots, (2\mathbf{1}_{\{y_n=y\}}-1)\varepsilon_n$
has the same distribution as $(\varepsilon_1, \varepsilon_2, \cdots, \varepsilon_n)$.

On the other hand, we also have

$$\mathbb{E}_{\boldsymbol{\varepsilon}}\left[\sup_{f\in\mathcal{F}}\left|\frac{1}{n}\sum_{i=1}^{n}\varepsilon_i\sup_{y'\neq y_i}[f(\mathbf{x}_i)]_{y'}\right|\right]$$

$$=\mathbb{E}_{\boldsymbol{\varepsilon}}\left[\sup_{f\in\mathcal{F}}\left|\frac{1}{n}\sum_{i=1}^{n}\varepsilon_i\sup_{y'\neq y_i}[f(\mathbf{x}_i)]_{y'}\sum_{y=1}^{M}\mathbf{1}_{\{y_i=y\}}\right|\right] \tag{171}$$

$$=\mathbb{E}_{\boldsymbol{\varepsilon}}\left[\sup_{f\in\mathcal{F}}\left|\frac{1}{n}\sum_{y=1}^{M}\sum_{i=1}^{n}\varepsilon_i\sup_{y'\neq y}[f(\mathbf{x}_i)]_{y'}\mathbf{1}_{\{y_i=y\}}\right|\right] \tag{172}$$

$$\leq\sum_{y=1}^{M}\mathbb{E}_{\boldsymbol{\varepsilon}}\left[\sup_{f\in\mathcal{F}}\left|\frac{1}{n}\sum_{i=1}^{n}\varepsilon_i\sup_{y'\neq y}[f(\mathbf{x}_i)]_{y'}\mathbf{1}_{\{y_i=y\}}\right|\right] \tag{173}$$

$$\leq\frac{1}{2}\sum_{y=1}^{M}\mathbb{E}_{\boldsymbol{\varepsilon}}\left[\sup_{f\in\mathcal{F}}\left|\frac{1}{n}\sum_{i=1}^{n}\varepsilon_i\sup_{y'\neq y}[f(\mathbf{x}_i)]_{y'}(2\mathbf{1}_{\{y_i=y\}}-1)\right|\right]$$

$$+\frac{1}{2}\sum_{y=1}^{M}\mathbb{E}_{\boldsymbol{\varepsilon}}\left[\sup_{f\in\mathcal{F}}\left|\frac{1}{n}\sum_{i=1}^{n}\varepsilon_i\sup_{y'\neq y}[f(\mathbf{x}_i)]_{y'}\right|\right] \tag{174}$$

$$=\frac{1}{2}\sum_{y=1}^{M}\mathbb{E}_{\boldsymbol{\varepsilon}}\left[\sup_{f\in\mathcal{F}}\left|\frac{1}{n}\sum_{i=1}^{n}\varepsilon_i\sup_{y'\neq y}[f(\mathbf{x}_i)]_{y'}\right|\right]$$

$$+\frac{1}{2}\sum_{y=1}^{M}\mathbb{E}_{\boldsymbol{\varepsilon}}\left[\sup_{f\in\mathcal{F}}\left|\frac{1}{n}\sum_{i=1}^{n}\varepsilon_i\sup_{y'\neq y}[f(\mathbf{x}_i)]_{y'}\right|\right] \tag{175}$$

$$=\sum_{y=1}^{M}\mathbb{E}_{\boldsymbol{\varepsilon}}\left[\sup_{f\in\mathcal{F}}\left|\frac{1}{n}\sum_{i=1}^{n}\varepsilon_i\sup_{y'\neq y}[f(\mathbf{x}_i)]_{y'}\right|\right], \tag{176}$$

where (175) follows from the fact that $(2\mathbf{1}_{\{y_1=y\}}-1)\varepsilon_1,(2\mathbf{1}_{\{y_2=y\}}-1)\varepsilon_2,\cdots,(2\mathbf{1}_{\{y_n=y\}}-1)\varepsilon_n)$ has the same distribution as $(\varepsilon_1,\varepsilon_2,\cdots,\varepsilon_n)$.

Now, for each fixed $y\in[M]$ and $M=2$, let $\hat{y}=[M]\setminus\{y\}$ we have

$$\mathbb{E}_{\boldsymbol{\varepsilon}}\left[\sup_{f\in\mathcal{F}}\left|\frac{1}{n}\sum_{i=1}^{n}\varepsilon_i\sup_{y'\neq y}[f(\mathbf{x}_i)]_{y'}\right|\right]$$

$$=\mathbb{E}_{\boldsymbol{\varepsilon}}\left[\sup_{f\in\mathcal{F}}\left|\frac{1}{n}\sum_{i=1}^{n}\varepsilon_i[f(\mathbf{x}_i)]_{\hat{y}}\right|\right] \tag{177}$$

$$\leq\mathbb{E}_{\boldsymbol{\varepsilon}}\left[\sup_{f\in\mathcal{F}}\left\|\frac{1}{n}\sum_{i=1}^{n}\varepsilon_if(\mathbf{x}_i)\right\|_{\infty}\right]. \tag{178}$$

It follows from (176) and (178) that

$$\mathbb{E}_{\boldsymbol{\varepsilon}}\left[\sup_{f\in\mathcal{F}}\left|\frac{1}{n}\sum_{i=1}^{n}\varepsilon_i\sup_{y'\neq y_i}[f(\mathbf{x}_i)]_{y'}\right|\right]$$

$$\leq M\mathbb{E}_{\boldsymbol{\varepsilon}}\left[\sup_{f\in\mathcal{F}}\left\|\frac{1}{n}\sum_{i=1}^{n}\varepsilon_if(\mathbf{x}_i)\right\|_{\infty}\right]. \tag{179}$$

From (162), (170), and (179), for $M=2$ we have

$$\mathbb{E}_{\boldsymbol{\varepsilon}}\left[\sup_{f\in\mathcal{F}}\left|\frac{1}{n}\sum_{i=1}^{n}\varepsilon_im_f(\mathbf{x}_i,y_i)\right|\right]\leq2M\mathbb{E}_{\boldsymbol{\varepsilon}}\left[\sup_{f\in\mathcal{F}}\left|\frac{1}{n}\sum_{i=1}^{n}\varepsilon_if(\mathbf{x}_i)\right|\right]. \tag{180}$$

## A.11 PROOF OF THEOREM 14

Let $(\mathbf{x}_1', y_1'), (\mathbf{x}_2', y_2'), \cdots, (\mathbf{x}_n', y_n')$ is an i.i.d. sequence with distribution $P_{XY}$ which is independent of $X^n Y^n$. Define

$$E(f) := \mathbb{E}_{\mathbf{X}'\mathbf{Y}'} \left[ \frac{1}{n} \sum_{i=1}^n \zeta(m_f(\mathbf{x}_i', y_i')) \right]. \tag{181}$$

Now, let $D = \{(\mathbf{x}_i, y_i) : i \in [n]\}$, and let $\tilde{D} = \{(\mathbf{x}_i, y_i) : i \in [n]\}$ be a set with only one sample different from $D$, i.e. the $k$-th sample is replaced by $(\tilde{\mathbf{x}}_k, \tilde{y}_k)$. Define

$$\hat{E}_D(f) := \frac{1}{n} \sum_{i=1}^n \zeta(m_f(\mathbf{x}_i, y_i)) \tag{182}$$

and

$$\Phi(D) := \sup_{f \in \mathcal{F}} E(f) - \hat{E}_D(f), \tag{183}$$

which is a function of $n$ independent random vectors $(\mathbf{x}_1, y_1), (\mathbf{x}_2, y_2), \cdots, (\mathbf{x}_n, y_n)$ where $(\mathbf{x}_i, y_i) \sim P_{XY}$ for all $i \in [n]$. Since $0 \le \zeta(x) \le 1$ for all $x \in \mathbb{R}$, from (181) and (182) we have

$$\left| \Phi(\tilde{D}) - \Phi(D) \right| \le \sup_{f \in \mathcal{F}} \frac{|\zeta(m_f(\mathbf{x}_k, y_k)) - \zeta(m_f(\tilde{\mathbf{x}}_k, \tilde{y}_k))|}{n} \tag{184}$$

$$\le \frac{1}{n}. \tag{185}$$

By McDiarmid's inequality Raginsky & Sason (2013), with probability at least $1 - \exp(-2t^2)$ we have

$$\sup_{f \in \mathcal{F}} \left( \frac{1}{n} \mathbb{E}_{\mathbf{X}'\mathbf{Y}'} \left[ \sum_{i=1}^n \zeta(m_f(\mathbf{x}_i', y_i')) \right] - \frac{1}{n} \sum_{i=1}^n \zeta(m_f(\mathbf{x}_i, y_i)) \right)$$
$$\le \mathbb{E}_{\mathbf{X}\mathbf{Y}} \left[ \sup_{f \in \mathcal{F}} \left( \mathbb{E}_{\mathbf{X}'\mathbf{Y}'} \left[ \frac{1}{n} \sum_{i=1}^n \zeta(m_f(\mathbf{x}_i', y_i')) \right] - \frac{1}{n} \sum_{i=1}^n \zeta(m_f(\mathbf{x}_i, y_i)) \right) \right] + \frac{t}{\sqrt{n}}. \tag{186}$$

Now, let $\bar{\zeta}(x) := \zeta(x) - \zeta(0)$, which is a $1/\gamma$-Lipschitz function with $\bar{\zeta}(0) = 0$. Then, we have

$$\mathbb{E}_{\mathbf{XY}}\left[\sup_{f\in\mathcal{F}}\left(\mathbb{E}_{\mathbf{X'Y'}}\left[\frac{1}{n}\sum_{i=1}^{n}\zeta(m_f(\mathbf{x}_i',y_i'))\right] - \frac{1}{n}\sum_{i=1}^{n}\zeta(m_f(\mathbf{x}_i,y_i))\right)\right] \tag{187}$$

$$\leq \mathbb{E}_{\mathbf{XY}}\left[\sup_{f\in\mathcal{F}}\left|\mathbb{E}_{\mathbf{X'Y'}}\left[\frac{1}{n}\sum_{i=1}^{n}\bar{\zeta}(m_f(\mathbf{x}_i',y_i'))\right] - \frac{1}{n}\sum_{i=1}^{n}\bar{\zeta}(m_f(\mathbf{x}_i,y_i))\right|\right] \tag{188}$$

$$= \mathbb{E}_{\mathbf{XY}}\left[\sup_{f\in\mathcal{F}}\left|\mathbb{E}_{\mathbf{X'Y'}}\left[\frac{1}{n}\sum_{i=1}^{n}\left(\bar{\zeta}(m_f(\mathbf{x}_i',y_i')) - \bar{\zeta}(m_f(\mathbf{x}_i,y_i))\right)\right]\right|\right] \tag{189}$$

$$\leq \mathbb{E}_{\mathbf{XY}}\left[\mathbb{E}_{\mathbf{X'Y'}}\left[\sup_{f\in\mathcal{F}}\left|\frac{1}{n}\sum_{i=1}^{n}\left(\bar{\zeta}(m_f(\mathbf{x}_i',y_i')) - \bar{\zeta}(m_f(\mathbf{x}_i,y_i))\right)\right|\right]\right] \tag{190}$$

$$\leq \frac{1}{\gamma}\mathbb{E}_{\mathbf{XY}}\left[\mathbb{E}_{\mathbf{X'Y'}}\left[\sup_{f\in\mathcal{F}}\left|\frac{1}{n}\sum_{i=1}^{n}\left(m_f(\mathbf{x}_i',y_i') - m_f(\mathbf{x}_i,y_i)\right)\right|\right]\right] \tag{191}$$

$$= \frac{1}{\gamma}\mathbb{E}_{\boldsymbol{\varepsilon}}\left[\mathbb{E}_{\mathbf{XY}}\left[\mathbb{E}_{\mathbf{X'Y'}}\left[\sup_{f\in\mathcal{F}}\left|\frac{1}{n}\sum_{i=1}^{n}\varepsilon_i\left(m_f(\mathbf{x}_i',y_i') - m_f(\mathbf{x}_i,y_i)\right)\right|\right]\right]\right] \tag{192}$$

$$\leq \frac{1}{\gamma}\mathbb{E}_{\boldsymbol{\varepsilon}}\left[\mathbb{E}_{\mathbf{X'Y'}}\left[\sup_{f\in\mathcal{F}}\left|\frac{1}{n}\sum_{i=1}^{n}\varepsilon_i m_f(\mathbf{x}_i',y_i')\right|\right]\right]$$

$$+ \frac{1}{\gamma}\mathbb{E}_{\boldsymbol{\varepsilon}}\left[\mathbb{E}_{\mathbf{XY}}\left[\sup_{f\in\mathcal{F}}\left|\frac{1}{n}\sum_{i=1}^{n}\varepsilon_i m_f(\mathbf{x}_i,y_i)\right|\right]\right] \tag{193}$$

$$= \frac{2}{\gamma}\mathbb{E}_{\boldsymbol{\varepsilon}}\left[\mathbb{E}_{\mathbf{XY}}\left[\sup_{f\in\mathcal{F}}\left|\frac{1}{n}\sum_{i=1}^{n}\varepsilon_i m_f(\mathbf{x}_i,y_i)\right|\right]\right] \tag{194}$$

$$= \frac{2}{\gamma}\mathbb{E}_{\mathbf{XY}}\left[\mathbb{E}_{\boldsymbol{\varepsilon}}\left[\sup_{f\in\mathcal{F}}\left|\frac{1}{n}\sum_{i=1}^{n}\varepsilon_i m_f(\mathbf{x}_i,y_i)\right|\right]\right] \tag{195}$$

$$\leq \frac{2\beta(M)}{\gamma}\mathbb{E}_{\mathbf{XY}}\left[\mathbb{E}_{\boldsymbol{\varepsilon}}\left[\sup_{f\in\mathcal{F}}\left\|\frac{1}{n}\sum_{i=1}^{n}\varepsilon_i f(\mathbf{x}_i)\right\|_{\infty}\right]\right] \tag{196}$$

where (192) follows from (Truong, 2022b, Lemma 25), and (196) follows from Lemma 13.

From (196), with probability at least $1 - \exp(-2t^2)$ we have

$$\sup_{f\in\mathcal{F}}\left(\mathbb{E}\left[\frac{1}{n}\sum_{i=1}^{n}\zeta(m_f(\mathbf{x}_i',y_i'))\right] - \frac{1}{n}\sum_{i=1}^{n}\zeta(m_f(\mathbf{x}_i,y_i))\right) \tag{197}$$

$$\leq \frac{2\beta(M)}{\gamma}\mathbb{E}\left[\sup_{f\in\mathcal{F}}\left\|\frac{1}{n}\sum_{i=1}^{n}\varepsilon_i f(\mathbf{x}_i)\right\|_{\infty}\right] + \frac{t}{\sqrt{n}}. \tag{198}$$

It follows that, with probability at least $1 - \exp(-2t^2)$,

$$\mathbb{E}_{\mathbf{X'},\mathbf{Y'}}\left[\frac{1}{n}\sum_{i=1}^{n}\zeta(m_f(\mathbf{x}_i',y_i'))\right] \leq \frac{1}{n}\sum_{i=1}^{n}\zeta(m_f(\mathbf{x}_i,y_i))$$

$$+ \frac{2\beta(M)}{\gamma}\mathbb{E}\left[\sup_{f\in\mathcal{F}}\left\|\frac{1}{n}\sum_{i=1}^{n}\varepsilon_i f(\mathbf{x}_i)\right\|_{\infty}\right] + \frac{t}{\sqrt{n}} \qquad \forall f\in\mathcal{F}, \tag{199}$$

or

$$\mathbb{E}[\zeta(m_f(\mathbf{x},y))] \leq \frac{1}{n}\sum_{i=1}^{n}\zeta(m_f(\mathbf{x}_i,y_i))$$

$$+ \frac{2\beta(M)}{\gamma}\mathbb{E}\left[\sup_{f\in\mathcal{F}}\left\|\frac{1}{n}\sum_{i=1}^{n}\varepsilon_i f(\mathbf{x}_i)\right\|_{\infty}\right] + \frac{t}{\sqrt{n}} \qquad \forall f\in\mathcal{F}. \tag{200}$$

Now, observe that

$$\mathbb{E}[\zeta(m_f(\mathbf{x}, y))]$$
$$= \mathbb{P}\big[m_f(\mathbf{x}, y) \leq 0\big] + \mathbb{E}[\zeta(m_f(\mathbf{x}, y))|0 \leq m_f(\mathbf{x}, y) \leq \gamma]\mathbb{P}[0 \leq m_f(\mathbf{x}, y) \leq \gamma] \quad (201)$$
$$\geq \mathbb{P}\big(m_f(\mathbf{x}, y) \leq 0\big). \quad (202)$$

From (200) and (202), with probability at least $1 - \exp(-2t^2)$,

$$\mathbb{P}\big[m_f(\mathbf{x}, y) \leq 0\big] \leq \frac{1}{n}\sum_{i=1}^{n}\zeta(m_f(\mathbf{x}_i, y_i))$$
$$+ \frac{2\beta(M)}{\gamma}\mathbb{E}\bigg[\sup_{f \in \mathcal{F}}\bigg\|\frac{1}{n}\sum_{i=1}^{n}\varepsilon_i f(\mathbf{x}_i)\bigg\|_\infty\bigg] + \frac{t}{\sqrt{n}} \quad \forall f \in \mathcal{F}. \quad (203)$$

Now, let $\gamma_k = 2^{-k}$ for all $k \in \mathbb{N}$. For any $\gamma \in (0, 1]$, there exists a $k \in \mathbb{N}$ such that $\gamma \in (\gamma_k, \gamma_{k-1}]$. Then, by applying (203) with $t$ being replaced by $t + \sqrt{\log k}$ and $\zeta(\cdot) = \zeta_k(\cdot)$ where

$$\zeta_k(x) := \begin{cases} 0, & \gamma_k \leq x \\ 1 - \frac{x}{\gamma_k} & 0 \leq x \leq \gamma_k \\ 1, & x \leq 0 \end{cases}, \quad (204)$$

with probability at least $1 - \exp(-2(t + \sqrt{\log k})^2)$, we have

$$\mathbb{P}\big[m_f(\mathbf{x}, y) \leq 0\big] \leq \frac{1}{n}\sum_{i=1}^{n}\zeta_k(m_f(\mathbf{x}_i, y_i))$$
$$+ \frac{2\beta(M)}{\gamma}\mathbb{E}\bigg[\sup_{f \in \mathcal{F}}\bigg\|\frac{1}{n}\sum_{i=1}^{n}\varepsilon_i f(\mathbf{x}_i)\bigg\|_\infty\bigg] + \frac{t + \sqrt{\log k}}{\sqrt{n}}, \quad \forall f \in \mathcal{F}. \quad (205)$$

By using the union bound, from (205), with probability at least $1 - \sum_{k \geq 1}\exp(-2(t + \sqrt{\log k})^2)$, it holds that

$$\mathbb{P}\big[m_f(\mathbf{x}, y) \leq 0\big] \leq \inf_{k \geq 1}\bigg[\frac{1}{n}\sum_{i=1}^{n}\zeta_k(m_f(\mathbf{x}_i, y_i))$$
$$+ \frac{2\beta(M)}{\gamma}\mathbb{E}\bigg[\sup_{f \in \mathcal{F}}\bigg\|\frac{1}{n}\sum_{i=1}^{n}\varepsilon_i f(\mathbf{x}_i)\bigg\|_\infty\bigg] + \frac{t + \sqrt{\log k}}{\sqrt{n}}\bigg], \quad \forall f \in \mathcal{F}. \quad (206)$$

On the other hand, it is easy to see that

$$\frac{1}{\gamma_k} \leq \frac{2}{\gamma}, \quad (207)$$

$$\frac{1}{n}\sum_{i=1}^{n}\zeta_k(m_f(\mathbf{x}_i, y_i)) \leq \frac{1}{n}\sum_{i=1}^{n}\zeta(m_f(\mathbf{x}_i, y_i)), \quad (208)$$

$$\sqrt{\log k} \leq \sqrt{\log\log_2\frac{1}{\gamma_k}} \leq \sqrt{\log\log_2\frac{2}{\gamma}}, \quad (209)$$

$$\sum_{k \geq 1}\exp(-2(t + \sqrt{\log k})^2) \leq \sum_{k \geq 1}k^2 e^{-2t^2} = \frac{\pi^2}{6}e^{-2t^2} \leq 2e^{-2t^2}. \quad (210)$$

Hence, by combining (207)–(210), and (206), with probability at least $1 - 2\exp(-2t^2)$, it holds that

$$\mathbb{P}\big[m_f(\mathbf{x}, y) \leq 0\big] \leq \inf_{\gamma \in (0, 1]}\bigg[\frac{1}{n}\sum_{i=1}^{n}\zeta(m_f(\mathbf{x}_i, y_i))$$
$$+ \frac{2\beta(M)}{\gamma}\mathbb{E}\bigg[\sup_{f \in \mathcal{F}}\bigg\|\frac{1}{n}\sum_{i=1}^{n}\varepsilon_i f(\mathbf{x}_i)\bigg\|_\infty\bigg] + \frac{t + \sqrt{\log\log_2(2\gamma^{-1})}}{\sqrt{n}}\bigg], \forall f \in \mathcal{F}. \quad (211)$$

From (211) we have

$$
\mathbb{P}\big[m_f(\mathbf{x}, y) \leq 0\big] \leq \inf_{\gamma \in (0,1]} \Bigg[ \frac{1}{n} \sum_{i=1}^{n} \zeta\big(m_f(\mathbf{x}_i, y_i)\big)
$$

$$
+ \frac{2\beta(M)}{\gamma} \mathbb{E}\bigg[ \sup_{f \in \mathcal{F}} \bigg\| \frac{1}{n} \sum_{i=1}^{n} \varepsilon_i f(\mathbf{x}_i) \bigg\|_{\infty} \bigg] + \frac{t + \sqrt{\log \log_2(2\gamma^{-1})}}{\sqrt{n}} \Bigg], \quad \forall f \in \mathcal{F}. \tag{212}
$$

This concludes our proof of Theorem 14.

## A.12    EXTRA NUMERICAL RESULTS

### A.12.1    EXPERIMENT 2

```python
model = keras.Sequential(
    [
        layers.Input(shape=input_shape),
        layers.Conv2D(32, kernel_size=(3, 3), activation="relu"),
        layers.AveragePooling2D(pool_size=(2, 2)),
        layers.Conv2D(64, kernel_size=(3, 3), activation="relu"),
        layers.AveragePooling2D(pool_size=(2, 2)),
        layers.Flatten(),
        layers.Dropout(0.5),
        layers.Dense(2, activation="sigmoid"),
    ]
)
model.summary()
```

Figure 2: CNN model with ReLU activations

In this experiment, we use a CNN (cf. Fig. 2) for classifying MNIST images (class 0 and class 1), i.e., $M = 2$, which consists of $n = 12665$ training examples.

For this model, we use ReLU for the first two convolutional layers, and the sigmoid $\sigma$ for the dense layer which satisfies $\sigma(x) - \sigma(0) = \frac{1}{2} \tanh\left(\frac{x}{2}\right)$ (an odd function with Lipschitz constant $1/4$).

Hence, by Theorem 10 and Lemma 17 it holds that $\mathcal{R}_n(\mathcal{F}) \leq F_3$, where

$$
F_3 \leq \underbrace{\frac{1}{4} \|\mathbf{W}\|_{\infty} F_2 + \frac{1}{2\sqrt{n}}}_{\text{Dense layer}}, \tag{213}
$$

$$
F_2 \leq \underbrace{\Bigg( \sup_{l \in [64]} \sum_{u=1}^{3} \sum_{v=1}^{3} \big| W_2^{(l)}(u, v) \big| \Bigg)}_{\text{The second convolutional layer}} F_1, \tag{214}
$$

$$
F_1 \leq \underbrace{\Bigg( \sup_{l \in [32]} \sum_{u=1}^{3} \sum_{v=1}^{3} \big| W_1^{(l)}(u, v) \big| \Bigg)}_{\text{The first convolutional layer}} F_0, \tag{215}
$$

$$
F_0 = \sqrt{\frac{d+1}{n}}. \tag{216}
$$

Numerical estimation of $F_3$ gives $\mathcal{R}_n(\mathcal{F}) \leq 0.0476$.

By Corollary 15 with probability at least $1 - \delta$, it holds that

$$
\mathbb{P}\big(m_f(\mathbf{x}, y) \leq 0\big) \leq \inf_{\gamma \in (0,1]} \left[ \frac{1}{n} \sum_{i=1}^{n} \zeta\big(m_f(\mathbf{x}_i, y_i)\big) \right.
$$

$$
\left. + \frac{4M}{\gamma} \mathcal{R}_n(\mathcal{F}) + \sqrt{\frac{\log \log_2(2\gamma^{-1})}{n}} + \sqrt{\frac{2}{n} \log \frac{3}{\delta}} \right] \tag{217}
$$

By setting $\delta = 5\%$, $\gamma = 1$, the generalisation error can be upper bounded by

$$
\mathbb{P}\big(m_f(\mathbf{x}, y) \leq 0\big) \leq 0.412806. \tag{218}
$$

For this model, the reported test error is $0.0009456$.

### A.12.2    EXPERIMENT 3

```python
model = keras.Sequential(
    [
        layers.Input(shape=input_shape),
        layers.Conv2D(32, kernel_size=(3, 3), activation="sigmoid"),
        layers.AveragePooling2D(pool_size=(2, 2)),
        layers.Conv2D(64, kernel_size=(3, 3), activation="sigmoid"),
        layers.AveragePooling2D(pool_size=(2, 2)),
        layers.Flatten(),
        layers.Dropout(0.5),
        layers.Dense(2, activation="softmax"),
    ]
)
model.summary()
```

Figure 3: CNN model with sigmoid activations

In this experiment, we use a CNN (cf. Fig. 3) for classifying MNIST images (class 0 and class 1), i.e., $M = 2$, which consists of $n = 12665$ training examples.

For this model, the sigmoid activation $\sigma$ satisfies $\sigma(x) - \sigma(0) = \frac{1}{2} \tanh\left(\frac{x}{2}\right)$ which is odd and has the Lipschitz constant $1/4$. In addition, for the dense layer, the sigmoid activation satisfies

$$
\big|\sigma(x) - \sigma(y)\big| \leq \frac{1}{4}|x - y|, \qquad \forall x, y \in \mathbb{R}. \tag{219}
$$

For this example, we assume that we compare the outputs at the layer right before the softmax layer to bound the generalisation error. Then, by Theorem 10 and Lemma 17 it holds that $\mathcal{R}_n(\mathcal{F}) \leq F_2$, where

$$
F_2 \leq \underbrace{\left( \frac{1}{4} \sup_{l \in [64]} \sum_{u=1}^{3} \sum_{v=1}^{3} \big|W_2^{(l)}(u, v)\big| \right) F_1 + \frac{1}{2\sqrt{n}}}_{\text{The second convolutional layer}}, \tag{220}
$$

$$
F_1 \leq \underbrace{\left( \frac{1}{4} \sup_{l \in [32]} \sum_{u=1}^{3} \sum_{v=1}^{3} \big|W_1^{(l)}(u, v)\big| \right) F_0 + \frac{1}{2\sqrt{n}}}_{\text{The first convolutional layer}}, \tag{221}
$$

$$
F_0 = \sqrt{\frac{d+1}{n}}. \tag{222}
$$

Numerical estimation of $F_2$ gives $\mathcal{R}_n(\mathcal{F}) \leq 0.03074$.

By Corollary 15 with probability at least $1 - \delta$, it holds that

$$\mathbb{P}\big(m_f(\mathbf{x}, y) \leq 0\big) \leq \inf_{\gamma \in (0,1]} \left[ \frac{1}{n} \sum_{i=1}^{n} \zeta\big(m_f(\mathbf{x}_i, y_i)\big) \right.$$

$$\left. + \frac{4M}{\gamma} \mathcal{R}_n(\mathcal{F}) + \sqrt{\frac{\log \log_2(2\gamma^{-1})}{n}} + \sqrt{\frac{2}{n} \log \frac{3}{\delta}} \right] \tag{223}$$

By setting $\delta = 5\%$, $\gamma = 1$, the generalisation error can be upper bounded by

$$\mathbb{P}\big(m_f(\mathbf{x}, y) \leq 0\big) \leq 0.2775. \tag{224}$$

For this model, the reported test error is $0.001418$.