# OpenReview forum: "On Rademacher Complexity-based Generalization Bounds for Deep Learning"
_ICLR.cc/2025/Conference — Submitted to ICLR 2025_

### Official Review · Reviewer_JAkr · 2024-10-22

**Soundness:** 3
**Presentation:** 3
**Contribution:** 3
**Rating:** 6
**Confidence:** 3

**Summary:**

This paper first generalizes the Talagrand contraction lemma to high-dimensional mapping case, then uses these new results to derive non-vacuous generalization bounds for CNNs. Their results show that the Rademacher complexity does not explicitly depend on the depth of the network for CNNs with some specific types of activation functions. some empirical results on the MNIST image classifications are also given.

**Strengths:**

1. This paper provides a novel and useful contraction lemma, which extends the previous Talagrand contraction lemma to high-dimensional mapping case.
2. Based on the new contraction lemma, the authors derive the bounds of the Rademacher complexity for CNNs, which does not explicitly depend on the depth of CNNs. This improves the results in previous papers.
3. The authors derive some non-vacuous generalization bounds for CNNs.

**Weaknesses:**

1. This paper only considers some specific types of activation functions.
2. The conclusion that the obtained Rademacher complexity does not explicitly depend on the depth of CNNs is not very convincing, since by Equations (30), (31), (34), the Rademacher complexity may exponentially depend on the depth of CNNs if the right-hand-side of (30), and (31) are large than 1. More discussions on this claim should be added.

**Questions:**

See the Weaknesses part. In line 251, "Let $\psi(x)$ is" should be "Let $\psi(x)$ be" .

**Details Of Ethics Concerns:**

No.

---

> ### Author Response · Authors · 2024-11-13
> **Reply to Reviewer's comments**
>
> Thank you much for your comments on our paper. The following are our answers to your questions:
>
> 1. This paper only considers some specific types of activation functions
>
> Answer:  Although our developed bounds are for any activation functions, but we think that they are only non-vacuous for CNNS with some special classes of activation functions. We believe that achieving non-vacuous bounds is a challenging task in deep neural networks in general.
>
> 2.  The conclusion that the obtained Rademacher complexity does not explicitly depend on the depth of CNNs is not very convincing, since by Equations (30), (31), (34), the Rademacher complexity may exponentially depend on the depth of CNNs if the right-hand-side of (30), and (31) are large than 1. More discussions on this claim should be added.
>
> Answer: In our work, by using Theorem 8 (or Theorem 2), we show that for CNNs or DNNs with some special classes of activation functions (ReLU, Leaky ReLU, PRLI, Sigmoid, and Tanh or odd functions in general), the Rademacher complexity can be upper bounded by
> $$
> O\bigg(\sqrt{\frac{1}{n}}\prod_{j=1}^L \mu_j M_{\infty} (j) \bigg),
> $$ where $M_{\infty} (j)$ is the infinity-norm of the weight matrix of the $j$-the layer, and $\mu_j$ is the Lipschitz constant of the activation in the $j$-the layer. Note that $|\sigma_j(0)|/\sqrt{n}$ is usually very small compared to $\mu_j M_{\infty} (j)$.
>
> Hence, under the assumption of a bounded product of the norms of weight matrices, our derived bound is independent of the network length. This represents a significant improvement over the bounds established by Golowich et al., particularly for deep neural networks (DNNs) with extensive network lengths. We have adopted the term "size-independent" from Golowich et al.'s research to convey this concept. Additionally, we highlight that our results apply to a broader range of activation functions, extending beyond the ReLU activations that were primarily considered in their work.
>
> We will ensure to add "under the assumption of a bounded product of the norms of weight matrices" before stating that "the bound is network-depth independent" in our revised version.
>
> Based on our contributions, we look forward to receiving your better score.

---

> > ### Comment · Reviewer_JAkr · 2024-11-22
> > **Thanks for the reply**
> >
> > Thanks for the reply. It partially solved my concerns. I would like to keep the score.

---

### Official Review · Reviewer_HHY8 · 2024-11-03

**Soundness:** 1
**Presentation:** 1
**Contribution:** 2
**Rating:** 3
**Confidence:** 4

**Summary:**

This work studies the Rademacher complexity for the class of CNNs using any of ReLU family activation functions. Based on the authors' new version of Talagrand's contraction lemma, they provide the upper bound on the Rademacher complexity. Also, they conducted numerical experiments verifying the gap between the upper bound and error.

**Strengths:**

They try to identify activation functions and consider wider hypothesis space (e.g. $\mathcal{H}_+$) to provide new contraction coefficients.

**Weaknesses:**

**Questionable significance**

Though this work provides the new contraction lemma, in my opinion, the result is not so significant.
Compared to the known vector-valued Talagrands' contraction lemma ([1, 2]), theorem 2 seems to be able to have poor upper bounds since there is an additional term $\frac{1}{\sqrt{n}}| \psi (0) |$, which can even worsen the upper bound since the denominator is sublinear.
To clarify, please add more details and rigorously compare the lemmas in [1] and [2].
(e.g. why considering $\mathcal{H}_+$ lead better upper bound, ...)


**Non-vacuous bound**

The authors claim that the result provides a non-vacuous bound for CNNs with a small number of classes.
To verify this, they conducted several experiments under various setups.
However, as illustrated in Nagarajan & Kolter (2019), the bound may be vacuous depending on the norms of matrices (e.g. kernel) even if we carefully choose a set of parameters.
Thus, to prove the tightness, please provide a theoretical analysis of how their bounds behave under different matrix norm conditions in light of the results from Nagarajan & Kolter (2019).

**Comparison with other CNN bound**

Though there are lines of research studying generalization bound for CNNs, the authors do not provide any comparison with other known bounds. To highlight the novelty of the work, kindly suggest including a specific comparison section in their paper, highlighting key differences between their approach and existing CNN generalization bounds.

**Clarity Concerns**

There is no proof sketch or idea provided within the main text for any of the theorems; full proof in the main text or deferred to the appendix. For better readability, kindly suggest that the authors include brief proof sketches or key insights for the main theorems in the main text while keeping full proofs in the appendix.



**Reference**

[1] Maurer (2016), A vector-contraction inequality for Rademacher complexities

[2] Foster and Rakhlin (2019), $\ell_\infty$ Vector Contraction for Rademacher Complexity

**Questions:**

**1.About the remark 3**

The second bullet in remark 3 states that theorem 2 improves the bound for the case $m=1$. However, in Lemma 1 (Talagrand's lemma for $m=1$), the contraction rate is $\mu$, not $2\mu$.


**2. depth-dependent**

In theorem 8, the upper bound $F_L$ of $\mathcal{R}_{n}( \mathcal{F} )$ is defined using the recursive relation (34).
Because of $\gamma_i$, $F_L$ seems depending on the depth, unlike the explanation in the remark 9. Could you provide a bit more details?

---

I am hoping that the authors will provide the clarifications stated therein in the rebuttal phase

---

> ### Author Response · Authors · 2024-11-13
> **Reply to reviewer's comments**
>
> Thank you very much for your comments on our paper.
>
> First, we provide rebuttals to your comments on the weaknesses of our paper.
>
> 1. Questionable significance:
>
> Answer: We believe that your comments on the significance of our paper is not correct. There are three reasons:
> + The contraction lemmas in Maurer (2016) and Foster and Rakhlin (2019) are different from our contraction lemma in Theorem 2. It does not have the same form as ours. We note that the contraction lemmas in Maurer (2016) and Foster and Rakhlin (2019) are used for mapping between $\mathbb{R}^m \to \mathbb{R}$, but our contraction lemmas are for the mapping $\mathbb{R}^m \to \mathbb{R}^m$.  In addition, as (cf. (1) in Maurer 2016):
> $$
> \mathbb{E}\_{\epsilon}\bigg[\sup_{f \in \mathcal{F}}\sum_{t=1}^n \epsilon_t \phi_t(f(x_t))\bigg]\leq \sqrt{2}\mu \mathbb{E}\_{\epsilon} \bigg[\sup_{f \in \mathcal{F}}\sum_{t=1}^n \sum_{i=1}^K \epsilon_{t,i} f_i(x_t)\bigg]\qquad (1),
> $$
> so the form of Rademacher complexity is changed between the RHS and LHS. Hence, this contraction lemma looks hard to apply for DNNs where we need to peel layer by layer.
> + At least for the ReLU family (ReLU,  Leaky ReLU, Parametric Rectifier Linear Unit), tanh, sin, or any other odd activation function, we have $\psi(0)=0$. Hence, by our Theorem 2 it holds that
> $$
> \mathbb{E}\_{\epsilon}\bigg[\sup_{h \in \mathcal{H}}\bigg\\|\frac{1}{n}\sum_{i=1}^n \epsilon_i \psi(h(\mathbf{x}\_i))\bigg\\|_{\infty}\bigg] \leq \mu \mathbb{E}\_{\epsilon}\bigg[\sup\_{h \in \mathcal{H}}\bigg\\| \sum\_{i=1}^n \epsilon_i h(\mathbf{x}\_i) \bigg\\|\_{\infty} \bigg],
> $$ which improves (1).  Note that this contraction lemma also improves Talagrand's results (see our comments in the following box).
>
> + As the peeling process, by using our Theorem 2 or Lemma 7, our Rademacher upper bound has the following form:
> $$
> O\bigg(\sqrt{\frac{1}{n}}\prod_{j=1}^L \mu_j M_{\infty} (j) \bigg)  \qquad (2),
> $$
> where $M_{\infty} (j)$ is the infinity-norm of the weight matrix of the $j$-the layer, and $\mu_j$ is the Lipschitz constant of the activation in the $j$-the layer since $|\sigma_j(0)|/\sqrt{n}$ is usually very small compared to $\mu_j M_{\infty} (j)$ for a large number of training examples. However, if we apply contraction lemmas in Maurer (2016) and Foster and Rakhlin (2019) for DNNs with $L$ layers, we may achieve the following upper bound (although we believe that it is hard to apply these lemmas for deep learning as mentioned above)
> $$
> O\bigg((\sqrt{2})^L \sqrt{\frac{1}{n}}\prod_{j=1}^L \mu_j M_{\infty} (j) \bigg).
> $$ This bound is inferior to Golowich et al. and our bounds since the constant term is exponential in $L$ (See a detailed discussion in  Golowich et al. (Size-independent sample complexity of neural networks)).
>
> 2. Non-vacuous bound:
>
> Answer: In this work, we choose the infinity-norm for developing our bound since we believe that this norm can lead to tight bounds (see our experiments). Choosing the right norm is a contribution of our paper. We note that Nagarajan and Kolter (2019) experimented on the bounds provided by Neyshabur et al. (2018) and Bartlett et al. (2017) for ReLU DNNs. However, these bounds are even inferior to Golowich et al.  (2018) (see a detailed discussions in Golowich et. al (Size-independent sample complexity of neural networks)). They also use different norms from ours. We believe that the main reason why these bounds are vacuous since they are not tight enough. Tightening Rademacher complexity bounds is our target in this work. We achieve this by developing a novel contraction lemma for mappings between vector spaces in Theorem 2.
>
> 3. Comparison with other CNN bounds:
>
> Answer: As discussed in Golowich et al. (Size-independent sample complexity of neural networks), their bound offers the best performance in terms of dependence on network depth. Therefore, in our paper, we primarily compare our results with those of Golowich et al. (see Section 6). For a more detailed comparison of Golowich et al.'s work with other existing bounds, readers are encouraged to refer directly to their paper. Additionally, our results are built upon a new contraction lemma for vector spaces (Theorem 2), which distinguishes our approach from previous methods. In response to your suggestions, we will incorporate these clarifications into the revised version of our paper to enhance its readability.
>
> 4. Clarity Concerns:
>
> Answer: In the main text, we already provided a proof for Theorem 8. The proofs of other lemmas are given in Appendix since they are quite long. As you can see in Appendix, the proof of Theorem 2 are based on many lemmas, hence it requires a lot of spaces even for only stating these lemmas.

---

> ### Author Response · Authors · 2024-11-13
> **Reply to Reviewer's comments (next).**
>
> Next, we provide answers to your questions:
>
> 1. About the remark 3:
> The second bullet in remark 3 states that theorem 2 improves the bound for the case $m=1$. However, in Lemma 1 (Talagrand's lemma for $m=1$), the contraction rate is $\mu$, (not $2\mu$).
>
> Answer: The Talagrand's contraction lemma in Lemma 1 is for
> $$
> \mathbb{E}_{\epsilon}\bigg[\sup\_{h \in \mathcal{H}} \sum\_{i=1}^n \epsilon\_i (\psi \circ h)(\mathbf{x}_i) \bigg].
> $$
>
> If the expression inside the bracket is absolute value, i.e.,
> $$
> \mathbb{E}_{\epsilon}\bigg[\sup\_{h \in \mathcal{H}} \bigg|\sum\_{i=1}^n \epsilon\_i (\psi \circ h)(\mathbf{x}_i)\bigg| \bigg],
> $$
>
> then it is well-known that (Ledoux & Talagrand, 1991, Theorem 4.12)
>
> $$
> \mathbb{E}_{\epsilon}\bigg[\sup\_{h \in \mathcal{H}} \bigg|\sum\_{i=1}^n \epsilon\_i (\psi \circ h)(\mathbf{x}_i)\bigg| \bigg]
> $$
>
> $$
> \leq 2 \mu \mathbb{E}_{\epsilon}\bigg[\sup\_{h \in \mathcal{H}} \bigg|\sum\_{i=1}^n \epsilon\_i h(\mathbf{x}_i)\bigg| \bigg] \qquad (3).
> $$
> Hence, our contraction lemma in Theorem 2 improves (3) for many special classes of activation functions (See our Theorem 2) since the contraction rate is only $\mu$.
>
> 2. In Theorem 8, the upper bound $F_L$ of $\mathcal{R}_n(\mathcal{F})$ is defined using the recursive relation (34). Because of $\gamma_i$, $F_L$ seems depending on the depth, unlike the explanation in the remark 9. Could you provide a bit more details?
>
> Answer: In our work, by using Theorem 8 (or Theorem 2), we show that for CNNs or DNNs with some special classes of activation functions (ReLU, Leaky ReLU, PRLI, Sigmoid, and Tanh or odd functions in general), the Rademacher complexity can be upper bounded by
> $$
> O\bigg(\sqrt{\frac{1}{n}}\prod_{j=1}^L \mu_j M_{\infty} (j) \bigg),
> $$ where $M_{\infty} (j)$ is the infinity-norm of the weight matrix of the $j$-the layer, and $\mu_j$ is the Lipschitz constant of the activation in the $j$-the layer. Note that $|\sigma_j(0)|/\sqrt{n}$ is usually very small compared to $\mu_j M_{\infty} (j)$.
>
> Hence, under the assumption of a bounded product of the norms of weight matrices, our derived bound is independent of the network length. This represents a significant improvement over the bounds established by Golowich et al., particularly for deep neural networks (DNNs) with extensive network lengths. We have adopted the term "size-independent" from Golowich et al.'s research to convey this concept. Additionally, we highlight that our results apply to a broader range of activation functions, extending beyond the ReLU activations that were primarily considered in their work.
>
> We will ensure to add "under the assumption of a bounded product of the norms of weight matrices" before stating that "the bound is network-depth independent" in our revised version.
>
> Based on our responses to your questions, we believe that your evaluation of our work is not correct. Our results are significant from both theoretical and practical perspectives. There are two main contributions in our work: (1) deriving a novel contraction lemma between vector spaces which improves Talagrand's contraction lemma for many cases; (2) deriving an upper bound on Rademacher complexity which improves Golowich et al.'s bounds and leads to non-vacuous generalisation error bound (at least) for CNNs with binary classification. To the best of our knowledge, this is the first result to show that the Rademacher complexity approach can yield meaningful (non-vacuous) generalization error bounds. This is particularly significant because our goal in developing generalization error bounds is to ensure they are applicable in practical settings. Therefore, we hope the reviewer will fairly evaluate our contribution.

---

> ### Comment · Reviewer_HHY8 · 2024-11-22
>
> Dear authors,
>
> Thank you for your thoughtful response. I appreciate the effort in addressing my concerns. However, I have some lingering questions:
>
> **About significance**
>
> As pointed out, the contraction lemmas in Maurer (2016) and Foster and Rakhlin (2019) apply to mappings between $\mathbb{R}^{m} \to \mathbb{R}$. In Theorem 1, the function $\psi$ is defined as a mapping from $\mathbb{R} \to \mathbb{R} $, serving as a coordinate-wise  function. Thus, i thought that they are comparable. I agree with that their contraction lemmas can be loose for specific case (e.g. ReLU family) since they consider more wide class of functions (i.e. $\psi$). However, i still believe that the effect of $\psi(0)/\sqrt{n}$ could be worse due to the recursion relation (34). I would appreciate it if you could explain more.
>
> **Non vacuous bound**
>
> As you mentioned, the product of matrix norms must be bounded to achieve a non-vacuous bound, which is generally not guaranteed. I find this assumption quite strong. Additionally, even if the product of the weight matrix norms is bounded, the Lipschitz constant still depends on the network depth, which remains a concern.
>
>
> **Comparision**
>
>
> Due to $\psi(0)/\sqrt{n}$, i am not sure your comparison in section 6 is valid for the case $\psi(0) \neq 0$. I would appreciate it if you could explain more.
>
>
> **Clarity**
>
> It seems there may have been a misunderstanding. My intention was to suggest that introducing the key proof sketch or central ideas in the main text would be preferable, with the full proof deferred to the Appendix.

---

> ### Author Response · Authors · 2024-11-22
>
> Thank you very much for your response to our rebuttal. We will answer your concerns question by question:
>
> 1. About significance:
>
> First, we mention why the contraction lemmas in Maurer (2016) and Foster and Rakhlin (2019) can not be applied to our problem in more details.
> + As we shown in Theorem 13 and Corollary 14, the generalisation error is bounded by some terms+ $\frac{2\beta(M)}{\gamma} \mathcal{R}_n(\mathcal{F})$, where
> $$
> \mathcal{R}_n(\mathcal{F}):=\mathbb{E}\_{\boldsymbol{\epsilon}}\bigg[ \sup\_{f \in \mathcal{F}}\bigg\\|\frac{1}{n}\sum\_{i=1}^n \epsilon\_i f(\mathbf{x}_i)\bigg\\|\_{\infty} \bigg]\qquad (4)
> $$ where $\beta(M)$ is some function of $M$.  The LHS of contraction lemmas in Maurer (2016) and Foster and Rakhlin (2019) don't have this form (cf. (1) above). Hence, they can not be applied to bound the Rademacher complexity in (4).
> + As we mentioned in the previous rebuttal, to bound the Rademacher complexity in DNNs, we need to strip layer by layer. This means that if the Rademacher complexity of layer $l+1$ has the form
> $$
> \mathbf{E}_{\boldsymbol{\epsilon}}\bigg[\sup\_{h \in \mathcal{H}} g\bigg(\frac{1}{n}\sum\_{i=1}^n \epsilon_i \psi(h(\mathbf{x}_i))\bigg)\bigg] \qquad \qquad (5),
> $$ then the Rademacher complexity of layer $l$ must have the form
> $$
> \mathbf{E}\_{\boldsymbol{\epsilon}}\bigg[\sup\_{h \in \mathcal{H}} g\bigg(\frac{1}{n}\sum\_{i=1}^n \epsilon\_i h(\mathbf{x}\_i)\bigg)\bigg]\qquad \qquad (6).
> $$
> However, it is easy to see that the RHS and LHS of (1) does not have this symmetric form. Hence, we believe that it is hard to apply them for DNNs. If you know any work which applied the contraction lemmas in Maurer (2016) and Foster and Rakhlin (2019) for DNNs, please let us know.
>
> Secondly, related to your concerns about the term $|\psi(0)|/\sqrt{n}$, our answer is as follows:
> + As we mentioned in Remark 3, for ReLU family activation functions (eg. ReLU, Leaky ReLU, Parametric ReLU) and odd activation functions (eg. tanh, sinh, sin), $\psi(0)=0$. Hence, we don't need to care about this term.
> + For the case $\psi(0)\neq 0$, you are right that we can not conclude anything for certain. However, $|\psi(0)|/\sqrt{n}$ is usually small compared to other terms. For example, if we use (34) in Theorem 8, then we have
> $$
> F_1=\sqrt{\frac{d+1}{n}}\mu_1 + \frac{|\sigma_1(0)|}{\sqrt{n}},
> $$ where $d$ is the image dimension (eg. $28 \times 28$ for MNIST) and $\mu_1$ is the Lipschitz constant of activation function for this layer.  It is easy to see that $\sqrt{d+1} \mu_1>> |\sigma_1(0)|$ in general.  Hence, the second term is usually negligible. In our simulation in Section 5, we obtain non-vacuous generalisation bound for sigmoid network by using the recursion in Theorem 8.
>
> 2. Non-vacuous bound:
> There is a misunderstanding here. We only mentioned that our Rademacher complexity does not depend on the network-depth under the condition that the product of matrix norms is bounded (eg. in Section 6). The purpose of this statement is to show that our Rademacher complexity bound can improve other bounds (eg. Golowich et al.) under the same condition.
>
> Our achieved non-vacuous bounds are based on our applications of Theorem 8 for different CNNs. It does not depend on the assumption that the product of matrix norms is bounded.
>
> 3. Comparison:
> As we mentioned in Section 6, we only mentioned that (68) in our paper holds for some classes of activation functions such as ReLU family and odd activation functions (since $\psi(0)=0$). We note that previous Rademacher complexity bounds (Golowich et al. (2018), Neyshabur et al. (2015), Bartlett et al. (2017), Neyshabur et al. (2018)) are mainly limited to ReLU DNNs.
>
> 4. Clarity:
> Actually, we already mentioned the main idea why we can achieve tighter contraction lemmas in the final note in Remark 3. This is also the main idea of proof.
>
> As we mentioned in the previous rebuttal, the proof is very long and based on many lemmas where we use different techniques for each lemmas. Based on your concern, we will add a short proof (The Proof of Lemma 16 for ReLU family) in our revised version. Hopefully, you find that this solution is acceptable.
>
> We hope that our answers resolve your concerns.

---

> > ### Comment · Reviewer_HHY8 · 2024-11-26
> >
> > Dear authors,
> >
> > I think the assumption is too strong and unrealistic in general. Also, I still believe readability can be improved. Thus, I would retain my original score.

---

> ### Author Response · Authors · 2024-11-26
>
> Could you please clarify which assumption you believe is too strong or unrealistic in general? We would like to clarify that the "bounded product of the norms of the weight matrices" is not a requirement for deriving any of the results in our paper; it was included solely for the purpose of comparison with Golowich et al.'s work. In response to this possible concern, we have completely removed this statement in the second revision of our paper.

---

> ### Comment · Reviewer_HHY8 · 2024-11-27
>
> Dear authors,
>
> Regarding "the bounded product of the norms of the weight matrices", I think it is quite restrictive even though it is the purpose of comparison. After your answer, I re-read Section 6, but I still found another unrealizable assumption: the weight matrices are sparse such that there is only one non-zero element. With this kind of assumption (e.g. $W$ contains only one non-zero row $\mathbf{a}$), we always have $ \lVert{W}\rVert_{F} = \lVert{\mathbf{a}}\rVert_2 \le \lVert{\mathbf{a}}\rVert_1 =\lVert{W}\rVert_{\infty}$. Thus, I would retain my original score.

---

> ### Author Response · Authors · 2024-11-27
>
> Thank you for your response.
>
> As per your concern, we have rewritten the comparison section to better clarify our approach. We hope that you are happy with this revised version. It is important to note that our goal is not to propose a bound that universally outperforms the one in Golowich et al. across all scenarios for general DNNs. Instead, our focus in this work is specifically on Convolutional Neural Networks (CNNs). The main contributions of our study are as follows:
>
> + We develop new contraction lemmas for high-dimensional mappings between vector spaces which extend and improve the Talagrand contraction lemma for many cases.
> + We apply our new contraction lemmas to each layer of a CNN.
> +  We validate our new theoretical results experimentally on CNNs for MNIST image classifications, and our bounds are non-vacuous when the number of classes is small.
>
> The bound proposed by Golowich et al., along with other related works, is primarily designed for general DNNs. In Section 6, we highlight the differences between our bound and theirs.

---

### Official Review · Reviewer_XMcb · 2024-11-03

**Soundness:** 2
**Presentation:** 1
**Contribution:** 2
**Rating:** 3
**Confidence:** 4

**Summary:**

This paper explores the theoretical underpinnings of generalization in deep learning, specifically focusing on Convolutional Neural Networks (CNNs). The authors propose a new approach using Rademacher complexity to derive non-vacuous generalisation bounds for CNNs with certain activation functions like ReLU, Leaky ReLU, and Sigmoid. These bounds, unlike prior results, do not explicitly depend on the network depth. The authors validate their findings through experiments on MNIST image classification.

**Strengths:**

The authors employ the Talagrand's contraction lemma to develop novel   Rademacher-complexity-based generalization bound for CNNs and specialze the bounds to some common types of activation functions.

**Weaknesses:**

1. The paper's maths and presentation need to be significantly improved.
2. The theoretical contributions seem not sound enough.

**Questions:**

1. Lemma 1: $\epsilon$ is not defined.
2. The definition of network length should be provided at the beginning of the paper. It actually refers to the number of layers in Section 6. It would be better to adopt the conventional name "network depth".
3. What is the benefit of the bound that does not depends on the network length? It could not reflect the effects of depth in deep learning.
4. In Theorem 8, $L$ is not defined. Futhremore, $F_L$ depends on $L$. Does it contradict with the authors' claim?
5. What do the numerical results in Section 5 imply? There is no elucidation. Futhremore, the authors only implement one experiment which is far from enough for verification.
6. The title is not accurate. This paper only studies CNNs.

---

> ### Author Response · Authors · 2024-11-13
> **Reply to reviewer's comments**
>
> Thank you very much for your comments on our paper.
>
> First, we answer your comments question by question:
> 1. Lemma 1: $\boldsymbol{\epsilon}$ is not defined.
>
> Answer: We think that the readers can understand what is $\boldsymbol{\epsilon}$ from the context. However, based on your concern, we will define $\boldsymbol{\epsilon}=(\epsilon_1,\epsilon_2,\cdots,\epsilon_n)$ in our revised version.
>
> 2. The definition of network length should be provided at the beginning of the paper. It actually refers to the number of layers in Section 6. It would be better to adopt the conventional name "network depth".
>
> Answer: We mentioned about $L$ in Section 3.1 when we first introduced CNN. We will mention that $L$ is called "network-length" in Section 3.1 in the revised version as your suggestion.
>
> 3. What is the benefit of the bound that does not depends on the network length? It could not reflect the effects of depth in deep learning.
>
> Answer: To answer this question, we should look at the best known bound by Theorem 1.2. in Golowich et al. (Size-independent sample complexity of neural networks, 2018) where the authors showed that the Rademacher complexity of DNNs with ReLU activation functions is upper bounded by
> $$
> \tilde{O}\bigg(\sqrt{\frac{L}{n}}\prod_{j=1}^L  M_F(j) \bigg).
> $$
> Here, $M_F(j)$ is the Frobenius norm of the weight matrix of the $j$-the layer, and $n$ is the number of training examples (Please also see Section 6 in our paper).
>
> In our work, by using Theorem 8 (or Theorem 2), we show that for CNNs or DNNs with some special classes of activation functions (ReLU, Leaky ReLU, PRLI, Sigmoid, and Tanh or odd functions in general), the Rademacher complexity can be upper bounded by
> $$
> O\bigg(\sqrt{\frac{1}{n}}\prod_{j=1}^L \mu_j M_{\infty} (j) \bigg),
> $$ where $M_{\infty} (j)$ is the infinity-norm of the weight matrix of the $j$-the layer, and $\mu_j$ is the Lipschitz constant of the activation in the $j$-the layer. Note that $|\sigma_j(0)|/\sqrt{n}$ is usually very small compared to $\mu_j M_{\infty} (j)$.
>
> Hence, under the assumption of a bounded product of the norms of weight matrices, our derived bound is independent of the network length. This represents a significant improvement over the bounds established by Golowich et al., particularly for deep neural networks (DNNs) with extensive network lengths. We have adopted the term "size-independent" from Golowich et al.'s research to convey this concept. Additionally, we highlight that our results apply to a broader range of activation functions, extending beyond the ReLU activations that were primarily considered in their work.
>
> We will ensure to add "under the assumption of a bounded product of the norms of weight matrices" before stating that "the bound is network-depth independent" in our revised version.
>
> 4. In Theorem 8, $L$ is not defined. Furthermore, $F_L$ depends on $L$. Does it contradict with the author's claim?
>
> Answer: In Theorem 8, we mention that "we consider the CNN defined in Section 3.1" where we define $L$.
> For the second question, please refer to our answer for Question 3.
>
> 5. What do the numerical results in Section 5 imply? There is no elucidation. Furthermore, the authors only implement one experiment which is far from enough for verification.
>
> Answer:  Our simulation results demonstrate that by refining the Rademacher complexity bound, we can derive non-vacuous generalization error estimates for certain CNN networks. To the best of our knowledge, this is the first result to show that the Rademacher complexity approach can yield meaningful (non-vacuous) generalization error bounds. This is particularly significant because our goal in developing generalization error bounds is to ensure they are applicable in practical settings.
>
> We also provided two extra experiments in Appendix A.12 and mentioned this in Section 5.
>
> 6. The title is not accurate. This paper only studies CNNs.
>
> Answer: Although we only limit our experiments on CNNs, however, our development results such as Theorem 2 and Lemma 7 can be applied for general DNNs. We can change the title to "On Rademacher Complexity-based Generalisation Bounds for CNNs" if necessary.
>
> Based on our responses to your questions, we believe that your evaluation of our work is not correct. Our results are significant from both theoretical and practical perspectives. There are two main contributions in our work: (1) deriving a novel contraction lemma between vector spaces which improves Talagrand's contraction lemma for many cases; (2) deriving an upper bound on Rademacher complexity which improves Golowich et al.'s bounds and leads to non-vacuous generalisation error bound (at least) for CNNs with binary classification. To the best of our knowledge, this is the first result to show that the Rademacher complexity approach can yield meaningful (non-vacuous) generalization error bounds. Therefore, we hope the reviewer will fairly evaluate our contribution.

---

> > ### Comment · Reviewer_XMcb · 2024-11-26
> >
> > Thank you for your response. I still have the following concerns.
> >
> > - In Lemma 1, the authors still has not addressed my question. I know that  $\varepsilon_i$'s are the Rademacher random variables, but they are not defined  until Eq. (16). I think this should not happen in a rigorous theoretical work. I also notice that in Eq. (4), the activation function $\sigma_i$ is  not defined as well.
> >
> > - I would suggest giving a title to Lemma 6 to make it consistent with Lemmas 5 and 7. Furthermore, I personally think that there should be some explanations follwing these theorems in Section 3.2. Now there is none.
> >
> > - It is unreasonable to claim that the Rademarcher complexity bound is independent of $L$ by assuming the product $\prod_{j=1}^L\mu_j M_{\infty}(j)$ is bounded. If the product is bounded, then $L$ is bounded too, which means the bound by Golowich et al. is also independent of $L$. The assumption is too strong and inconvincing.
> >
> > - The numerical results only consider a 3-layer CNN and so the size of the CNN is much smaller than the number of training data. That is why the generalization bound has a small value. As the number of layers increas, the bound will likely to be vacuous again.
> >
> > - I still believe the presentation of the paper could be further improved and is not yet ready for publication. Currently, the paper primarily stacks a series of theorems and lemmas, lacking sufficient insightful explanations to guide the readers. Furthermore, a considerable amount of space is taken up by lengthy multiline equations and extended proofs, some of which could be streamlined to enhance clarity and focus.

---

> ### Author Response · Authors · 2024-11-26
>
> Thank you very much for your responses. The following are our answers to your concerns:
>
> + In Lemma 1, the authors still has not addressed my question. I know that
> $\epsilon_i$'s are the Rademacher random variables, but they are not defined until Eq. (16). I think this should not happen in a rigorous theoretical work. I also notice that in Eq. (4), the activation function
>  is not defined as well.
>
> Answer: Thank you for pointing our this. We already defined this notation in our second revised version.
>
> + I would suggest giving a title to Lemma 6 to make it consistent with Lemmas 5 and 7. Furthermore, I personally think that there should be some explanations following these theorems in Section 3.2. Now there is none.
>
> Answer: As per your suggestions, we added a title called "Dropout Layers" for this lemma. In addition, we added a remark at the end of Section 3.2 to show the main ideas how to achieve results in Lemmas 6, 7, and 8.
>
> + It is unreasonable to claim that the Rademarcher complexity bound is independent of $L$ by assuming the product $\prod_{j=1}^L \mu_j M_{\infty}(j)$ is bounded. If the product is bounded, then $L$
>  is bounded too, which means the bound by Golowich et al. is also independent of $L$
> . The assumption is too strong and inconvincing.
>
> Answer: We have completely removed the statement "our Rademacher complexity is explicitly independent of the network depth" in our second revision. This statement, which was originally used in Golowich et al. (2018, 2020) in their work "Size-independent sample complexity of neural networks", was removed in response to your concerns.
>
> + The numerical results only consider a 3-layer CNN and so the size of the CNN is much smaller than the number of training data. That is why the generalization bound has a small value. As the number of layers increase, the bound will likely to be vacuous again.
>
> Answer: We believe that the relatively small size of a CNN compared to the number of training samples $n$ does not necessarily imply a small generalization error bound. This is because the norm of the weights $\\|\mathbf{W}_j\\|\_{\infty}$ (after training) does depend on $n$ for all $j \in [L]$.
>
> On the other hand, in this work, our goal is not to claim that our bounds are tight in every scenario. Rather, we aim to demonstrate that by refining the Rademacher complexity bounds, we can derive non-vacuous results. Achieving tight generalization error bounds is a complex and gradual process, and we hope that our contributions in this research topic will be recognized as valuable steps forward. We look forward to your acknowledgment of the progress we have made in this research direction.
>
> +  I still believe the presentation of the paper could be further improved and is not yet ready for publication. Currently, the paper primarily stacks a series of theorems and lemmas, lacking sufficient insightful explanations to guide the readers. Furthermore, a considerable amount of space is taken up by lengthy multiline equations and extended proofs, some of which could be streamlined to enhance clarity and focus.
>
> Answer: In our revisions, we have made every effort to improve the presentation in response to the reviewers' concerns, including adding additional insights. However, since this is a theoretical work and many of the theorems rely on mathematical techniques, removing these details would make the arguments difficult to follow. As such, we have retained them in the Appendix for clarity and completeness.

---

> > ### Comment · Reviewer_XMcb · 2024-12-02
> >
> > Thank you for your response. After careful consideration, I think that the current manuscript still lacks significant contributions. I would like to maintain my score.

---

### Author Response · Authors · 2024-11-20

Dear Reviewers,

Based on your suggestions for presentation, we have made improvements in the revised version. Specifically:

+ We provide a clearer explanation in Section 6 regarding why our Rademacher complexity bound improves previous works.
+ We have revised the presentation of Remark 3, where we clarify why our contraction lemma in Theorem 2 provides an improvement over Talagrand's contraction lemma in many cases.

Looking forward to receiving your re-considerations.

---

### Author Response · Authors · 2024-11-26

Dear Reviewers,

In this second revised version, we have made several improvements to the presentation. Specifically, the key changes are:

+ We have completely removed the statement "our Rademacher complexity is explicitly independent of the network depth." This statement, which was originally used in Golowich et al. (2018, 2020) in their work "Size-independent sample complexity of neural networks", was removed in response to reviewers' concerns.

+ We have rewritten the comparison section (Section 6), where we highlight how our results differ from previous ones, including those from Golowich et al.

Looking forward to receiving your positive feedbacks.

---

### Meta-Review · Area_Chair_6A1W · 2024-12-23

**Metareview:**

This paper presents contraction lemmas for high-dimensional mappings between vector spaces, extending Talagrand's contraction lemma, and applies these to derive Rademacher complexity bounds for Convolutional Neural Networks (CNNs). The authors claim to achieve non-vacuous generalization bounds for CNNs on MNIST classification with a small number of classes. While the paper attempts to extend existing contraction lemmas and improve Rademacher complexity bounds for CNNs, it suffers from significant limitations. The results primarily apply to specific activation functions, and the claims rely on assumptions about weight matrix norms that reviewers found unrealistic. The presentation lacks clarity, intuitive explanations, and clear comparisons to existing work. Moreover, the practical significance of the bounds is not convincingly demonstrated.

The paper's theoretical contributions are undermined by these limitations in scope and assumptions. The initial claim about depth-independence, which was later removed, raised serious concerns about the validity of the results. The lack of clear comparisons to existing bounds and persistent clarity issues make it difficult to assess the impact of the work. Given these shortcomings, the paper does not meet the standard for publication at this time and requires substantial revision and development.

**Additional Comments On Reviewer Discussion:**

The discussion revealed several critical issues with the paper. Reviewers strongly questioned the depth-independence claim, noting it relied on unrealistic assumptions. Although the authors removed this claim in revisions, this raised concerns about the paper's original formulation. Reviewers also requested clearer comparisons to other CNN bounds, which the authors attempted to address but reviewers still found insufficient. Significant issues with clarity and presentation persisted despite some improvements by the authors. Moreover, reviewers questioned the practical significance of the bounds given the limitations on activation functions and assumptions made.

While the authors attempted to address these points through revisions, most reviewers remained unconvinced that the fundamental issues were resolved. The need for substantial revisions to key claims, combined with ongoing clarity issues and questions about practical significance, led to the decision to reject the paper. The work requires significant further development before it can be considered for publication.

---

### Decision · Program_Chairs · 2025-01-22

Reject